# Blood pro-resolving mediators are linked with synovial pathology and are predictive of DMARD responsiveness in rheumatoid arthritis

Esteban A. Gomez [1,5], Romain A. Colas[1,5], Patricia R. Souza [1,5], Rebecca Hands[2], Myles J. Lewis [2], Conrad Bessant [3], Costantino Pitzalis [2,4,6] & Jesmond Dalli [1,4,6✉]

Biomarkers are needed for predicting the effectiveness of disease modifying antirheumatic drugs (DMARDs). Here, using functional lipid mediator profiling and deeply phenotyped patients with early rheumatoid arthritis (RA), we observe that peripheral blood specialized pro-resolving mediator (SPM) concentrations are linked with both DMARD responsiveness and disease pathotype. Machine learning analysis demonstrates that baseline plasma concentrations of resolvin D4, 10S, 17S-dihydroxy-docosapentaenoic acid, 15R-Lipoxin (LX)A$_4$ and n-3 docosapentaenoic-derived Maresin 1 are predictive of DMARD responsiveness at 6 months. Assessment of circulating SPM concentrations 6-months after treatment initiation establishes that differences between responders and non-responders are maintained, with a decrease in SPM concentrations in patients resistant to DMARD therapy. These findings elucidate the potential utility of plasma SPM concentrations as biomarkers of DMARD responsiveness in RA.

[1] William Harvey Research Institute, Barts and The London School of Medicine and Dentistry, Queen Mary University of London, Charterhouse Square, London EC1M 6BQ, UK. [2] Centre for Experimental Medicine and Rheumatology, William Harvey Research Institute, Barts and The London School of Medicine and Dentistry, Queen Mary University of London, London EC1M 6BQ, UK. [3] School of Biological and Chemical Sciences, Queen Mary University of London, Mile End Road, London E1 4NS, UK. [4] Centre for Inflammation and Therapeutic Innovation, Queen Mary University of London, London EC1M 6BQ, UK. [5] These authors contributed equally: Esteban A. Gomez, Romain A. Colas, Patricia R. Souza. [6] These authors jointly supervised this work: Costantino Pitzalis, Jesmond Dalli. ✉email: j.dalli@qmul.ac.uk

Rheumatoid arthritis (RA) is characterized by unremitting joint inflammation that results in bone and cartilage destruction and a decreased quality of life. Disease-modifying anti-rheumatic drugs (DMARDs) are widely used as a front line therapeutic in the treatment of RA. Here, low dose methotrexate (MTX) is the anchor drug, where it is administered alone or in combination with other DMARDs such as hydroxychloroquine and sulfasalazine. However, patients treated with DMARDs rarely go into full remission, with as many as 50% of patients being resistant to DMARD treatment or developing resistance over time[1]. In addition, DMARDs exert several unwanted side effects including an increased risk of infection[2] and liver function abnormalities[3,4].

Studies in early RA patients demonstrate that this condition presents with different synovial molecular and histological features that display distinct responsiveness to DMARD therapy[5]. These observations suggest that the ability of DMARDs to limit joint inflammation relies on regulating host protective pathways that may become dysregulated in distinct pathotypes. Amongst the DMARDs, MTX is administered to the majority (>80%) of RA patients. Several mechanisms of action have been proposed for the observed beneficial effects of low dose MTX in RA[6]. Amongst these is the depletion of purine and thymidine pools, reducing cellular proliferation and promoting apoptosis of mitogenically stimulated cells[7]. Furthermore, a CD39 expression in peripheral blood regulatory T cells is linked with the observed beneficial actions of MTX, whereby, patients that displayed a lower density of this receptor on peripheral blood regulatory T cells were unresponsive to MTX[8]. Given the wide range of unwanted side effects and the large number of patients unresponsive to DMARD treatment (~50%)[1], there is great interest in identifying predictive biomarkers. This is because such biomarkers are anticipated to reduce the unnecessary exposure of patients that are unlikely to respond to this class of drugs to their negative side effects. They will also provide early access to more effective therapeutics thereby reducing disease progression.

It is now appreciated that RA may arise from a decreased ability of the host immune response to engage resolution programmes that prevent the precipitation of acute inflammation into chronicity[9–11]. It is now well appreciated that central to the termination of ongoing inflammation is a newly uncovered genus of mediators. These molecules, termed as specialized pro-resolving mediators (SPM), are produced by immune cells via the enzymatic conversion of essential fatty acids, including the omega-3 fatty acids n-3 docosapentaenoic acid (n-3 DPA) and docosahexaenoic acid (DHA). These mediators carry distinct stereochemistries that were established using a matching approach[12]. SPM regulate both innate and adaptive immune responses and their production is reflective of the activation status of different immune cells[13–15]. Results obtained using experimental systems demonstrate that during delayed or non-resolving joint inflammation there is a downregulation of several SPM including the DHA-derived resolvin (Rv) D3[16]. In arthritic patients, synovial levels of the eicosapentaenoic acid (EPA)-derived RvE2 were found to correlate with decreased joint pain[11]. Furthermore, strategies to increase the production of these molecules through essential fatty acid supplementation or administration of the mediators themselves are linked with decreased joint inflammation and promotion of joint protection[17–19].

Therefore, in the present study we questioned whether the protective actions of MTX relied, at least in part, on the regulation of these endogenous protective pathways and whether endogenous SPM levels were predictive of responsiveness to MTX mono or co-therapy. Using plasma from deeply characterized early-arthritis patients[5] collected prior to treatment initiation we observe a segregation in lipid mediator profiles between those patients that responded to DMARDs and those that did not. Furthermore, plasma SPM concentrations are also diagnostic of disease pathotype. Difference in peripheral blood pro-resolving lipid mediators in DMARD responders and non-responders persist up to 6 months post DMARD initiation. Together these findings suggest that plasma SPM concentrations are characteristic of both treatment responsiveness and disease pathotypes in RA.

## Results

**Lipid mediator concentrations are predictive of responsiveness to DMARDs.** In order to determine whether peripheral blood SPM concentrations are predictive of DMARD responsiveness in patients with RA, we investigated plasma lipid mediator profiles in matched, deeply phenotyped early RA patients prior to treatment initiation (see Supplementary Table 1 for patient information). Plasma lipid mediators were identified in accordance with published criteria[20] that include matching of the retention time in liquid chromatography and at least six diagnostic ions in the tandem mass spectrum. In RA patient plasma, we identified mediators from all four major essential fatty acid metabolomes, i.e. arachidonic acid (AA), EPA, n-3 DPA and DHA (Supplementary Tables 2 and 3). These included the EPA, n-3 DPA and DHA-derived resolvins and the n-3 DPA and DHA-derived protectins and maresins (Supplementary Figs. 1–4 and Supplementary Tables 2 and 3). We next used orthogonal projections to latent structures discriminant analysis (OPLS-DA), which generates a regression model based on concentrations of lipid mediators differently expressed between two groups[21], to assess the concentrations of identified mediators between DMARD responders and DMARD non-responders. Here we observed two distinct clusters representing each of these patient groups (Fig. 1a, b). Since circulating peripheral blood cells are significant contributors to plasma lipid mediator profiles we next assessed whether there were differences between peripheral blood cell counts in these two patient groups. This analysis revealed that circulating platelet and phagocyte counts were essentially identical in the two groups (Supplementary Fig. 5).

We next used the machine-learning method *random forests* to build models based on plasma lipid mediator concentrations to further evaluate whether pre-treatment levels of these mediators were linked with DMARD responsiveness. Using this approach, we first assessed whether specific lipid mediator metabolomes were predictive of treatment responsiveness using plasma lipid mediator profiles from 30 DMARD responders and 22 DMARD non-responders. Here we found that cumulative concentrations of the DHA (that includes the D-series resolvins, protectins and maresins) and n-3 DPA (that includes the 13-series resolvins, D-series resolvins, protectins and maresins) metabolomes were the most accurate at predicting whether a patient would respond to treatment or not. The accuracy for the DHA metabolome at predicting outcome was of ~81% and that of the n-3 DPA metabolome was of ~69% (Fig. 1c and Table 1). Of note, these values were also higher than those obtained using a combination of clinical parameters that included the DAS28-ESR and rheumatoid factor concentrations (Fig. 1c and Table 1). In order to validate the robustness of our model we obtained peripheral blood lipid mediator profiles from a second cohort of DMARD naive patients composed of 36 responders and 22 non-responders, and tested whether the models generated using the different mediator metabolomes predicted outcome for patients in this cohort (see Supplementary Tables 4 and 5 for patient clinical parameters and lipid mediator concentrations). For this purpose, we assessed the receiver operating characteristic (ROC)

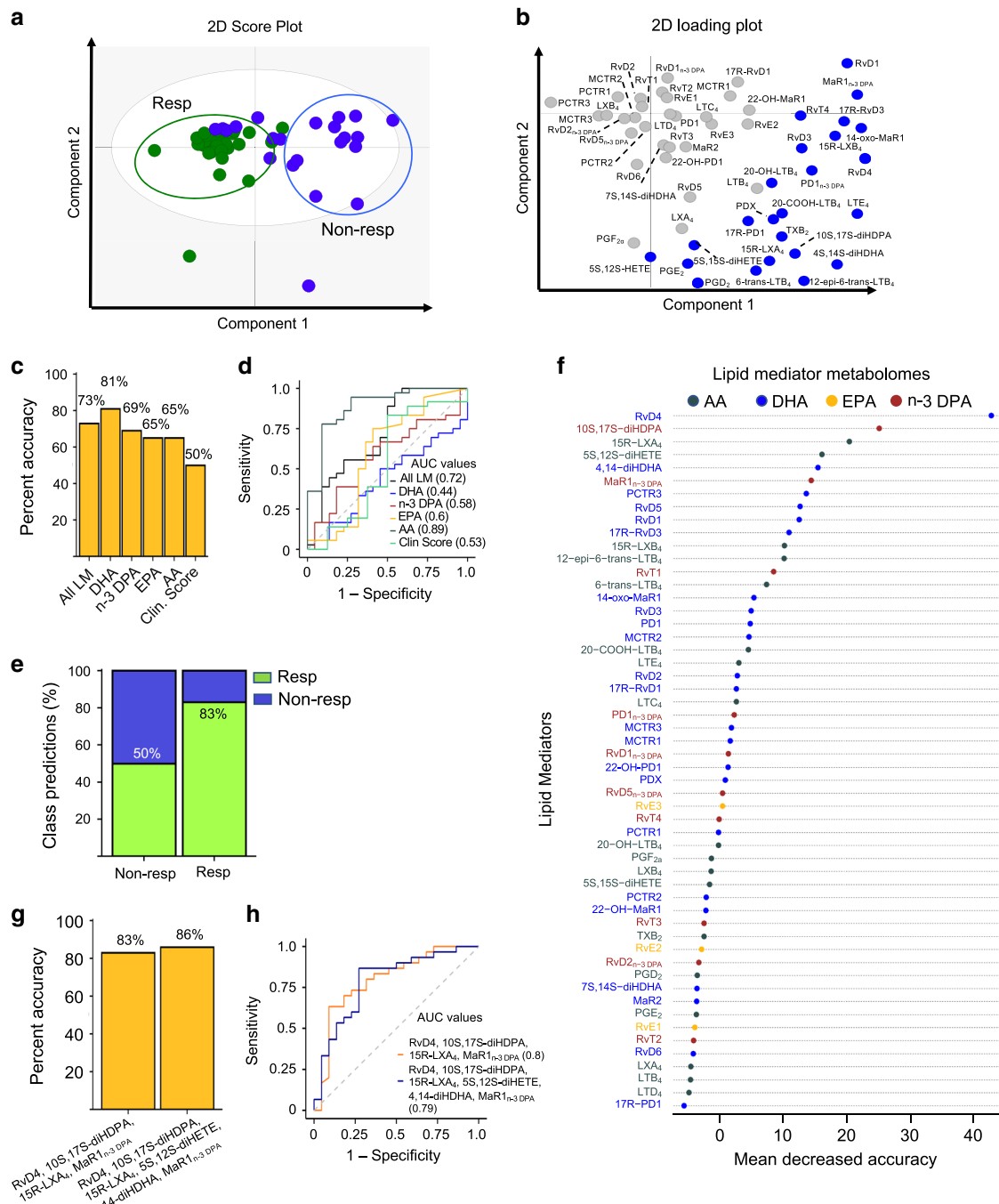

**Fig. 1 Baseline lipid mediator profiles are predictive of DMARD responsiveness in RA patients.** Plasma was collected from RA patients prior to the initiation of treatment with DMARDs and lipid mediator concentrations established using LC-MS/MS-based lipid mediator profiling (see 'Methods' for details). **a**, **b** OPLS-DA analysis of peripheral blood lipid mediator concentrations for DMARD responders (Resp) and DMARD non-responders (Non-Resp). **a** Two-dimensional score plot. Grey circle represents the 95% confidence regions. **b** Two-dimensional loading plots. Lipid mediators with VIP score >1 are upregulated in Non-Resp and denoted in blue. Results are representative of $n = 30$ Resp and $n = 22$ Non-Resp. **c** Percent accuracy score of prediction models based on the combination of all lipid mediators identified and quantified (AL LM) or individual fatty acid metabolomes as indicated. Clin. Score = clinical score (see 'Methods' for parameters included). **d** ROC curves and AUC values (provided in brackets) for predictive models. **e** Classification predictions for each class (sensitivity and specificity) of the n-3 DPA model. Green indicates the samples that were predicted as Resp while blue indicates those patients predicted Non-Resp. Percentages indicate true positives (Resp class) and true negatives (Non-Resp class). **f** Relevance of lipid mediators in the prediction performance of the "ALL LM" model based on decreasing accuracy. **g** Percent accuracy score of models using the indicated SPM. **h** ROC curves and AUC values for predictive models based on the indicated SPM. All the models were created using the random forest methodology ("randomForest" package from R). Source data are provided as a Source data file.

**Table 1 Summary of prediction models created using support vector machine and random forest.**

| Machine learning methodology | Samples | Model | Number of variables | % Accuracy score | Model validation | | | | | | Model evaluation | | | | | | |
|---|---|---|---|---|---|---|---|---|---|---|---|---|---|---|---|---|---|
| | | | | | TPR | TNR | TP | FP | TN | FN | TPR | TNR | TP | FP | TN | FN | AUC |
| randomForest (RF) | All samples (1st cohort) | Four metabolomes | 54 | 73 | 0.87 | 0.55 | 87 | 45 | 55 | 13 | 0.58 | 0.64 | 58 | 36 | 64 | 42 | 0.72 |
| randomForest (RF) | All samples (1st cohort) | DHA metabolome | 23 | 81 | 0.9 | 0.68 | 90 | 32 | 68 | 10 | 0.39 | 0.55 | 39 | 45 | 55 | 61 | 0.44 |
| randomForest (RF) | All samples (1st cohort) | n-3 DPA metabolome | 10 | 69 | 0.83 | 0.5 | 83 | 50 | 50 | 17 | 0.78 | 0.32 | 78 | 68 | 32 | 22 | 0.58 |
| randomForest (RF) | All samples (1st cohort) | EPA metabolome | 3 | 65 | 0.8 | 0.45 | 80 | 55 | 45 | 20 | 0.53 | 0.32 | 53 | 68 | 32 | 47 | 0.6 |
| randomForest (RF) | All samples (1st cohort) | AA metabolome | 18 | 65 | 0.77 | 0.5 | 77 | 50 | 50 | 23 | 0.83 | 0.77 | 83 | 23 | 77 | 17 | 0.89 |
| randomForest (RF) | All samples (1st cohort) | Clin. Score | 11 | 50 | 0.6 | 0.36 | 60 | 64 | 36 | 40 | 0.03 | 0.88 | 3 | 12 | 88 | 97 | 0.53 |
| randomForest (RF) | All samples (2nd cohort) | RvD4, 10S,17S-diHDPA, 15R-LXA$_4$, MaRI$_{n-3\ DPA}$ | 4 | 83 | 0.92 | 0.68 | 92 | 32 | 68 | 8 | 0.83 | 0.59 | 83 | 41 | 59 | 17 | 0.8 |
| randomForest (RF) | All samples (2nd cohort) | RvD4, 10S,17S-diHDPA, 15R-LXA$_4$, 5S,12S-diHETE, 4S,14S-diHDHA, MaRI$_{n-3\ DPA}$ | 6 | 86 | 0.94 | 0.73 | 94 | 27 | 73 | 6 | 0.87 | 0.64 | 87 | 36 | 64 | 13 | 0.79 |
| randomForest (RF) | Fibroid samples (1st & 2nd cohort) | RvD4, 10S,17S-diHDPA, 15R-LXA$_4$, MaRI$_{n-3\ DPA}$ | 4 | 88 | 0.89 | 0.87 | 89 | 13 | 87 | 11 | N/A | N/A | N/A | N/A | N/A | N/A | N/A |
| randomForest (RF) | Lymphoid samples (1st & 2nd cohort) | RvD4, 10S,17S-diHDPA, 15R-LXA$_4$, MaRI$_{n-3\ DPA}$ | 4 | 83 | 0.89 | 0.7 | 89 | 30 | 70 | 11 | N/A | N/A | N/A | N/A | N/A | N/A | N/A |
| randomForest (RF) | Myeloid samples (1st & 2nd cohort) | RvD4, 10S,17S-diHDPA, 15R-LXA$_4$, MaRI$_{n-3\ DPA}$ | 4 | 70 | 0.86 | 0.47 | 86 | 53 | 47 | 14 | N/A | N/A | N/A | N/A | N/A | N/A | N/A |
| Classyfire (SVM) | All samples (1st cohort) | Four metabolomes | 54 | 61 | 0.63 | 0.54 | 63 | 46 | 54 | 37 | 0.94 | 0.05 | 94 | 95 | 5 | 6 | 0.53 |
| Classyfire (SVM) | All samples (1st cohort) | DHA metabolome | 23 | 62 | 0.65 | 0.56 | 65 | 44 | 56 | 35 | 0.78 | 0.18 | 78 | 82 | 18 | 22 | 0.54 |
| Classyfire (SVM) | All samples (1st cohort) | n-3 DPA metabolome | 10 | 61 | 0.63 | 0.54 | 63 | 46 | 54 | 37 | 0.94 | 0.23 | 94 | 77 | 23 | 6 | 0.66 |
| Classyfire (SVM) | All samples (1st cohort) | EPA metabolome | 3 | 60 | 0.62 | 0.52 | 62 | 48 | 52 | 38 | 0.92 | 0.05 | 92 | 95 | 5 | 8 | 0.58 |
| Classyfire (SVM) | All samples (1st cohort) | AA metabolome | 18 | 58 | 0.6 | 0.48 | 60 | 52 | 48 | 40 | 0.92 | 0.09 | 92 | 91 | 9 | 8 | 0.66 |

TPR sensitivity, TNR specificity, TP true positives, FP false positives, TN true negatives, FN false negatives, AUC area under the curve.

curve, which evaluates the diagnostic potential of a classifier by varying its discrimination threshold. Assessment of the area under the ROC curve demonstrated that the DHA metabolome gave an AUC of 0.44, whereas the n-3 DPA metabolome gave an AUC of 0.58 (Fig. 1d, Supplementary Fig. 6 and Table 1). Similar findings were made using support vector machines, a different machine-learning strategy. Here, the DHA metabolome gave the highest accuracy score of ~62%, an AUC of 0.54, while the n-3 DPA metabolome gave an accuracy score of 61%, an AUC of 0.66 (Table 1). We further evaluated the ability of the n-3 DPA metabolome-based model to accurately categorize patients using the resulting confusion matrix of the model. Here we found that the model based on concentrations of n-3 DPA-derived mediators was able to correctly classify ~83% of responders in the appropriate category (Fig. 1e). Thus, these results indicate that baseline peripheral blood lipid mediator profiles are linked with DMARD treatment outcome.

**Identification of specific SPM that are predictive of treatment outcome.** Having found that lipid mediator profiles are linked with responsiveness to DMARD treatment in RA patients we next investigated whether we could identify specific lipid mediators that may be useful as biomarkers for treatment responsiveness. For this purpose, we conducted a random forest "importance" analysis that identifies the relevance of every mediator in the performance of the model based on the prediction accuracy. Here we found that the DHA-derived RvD4 (4S,5R,17S-trihydroxy-6E,8E,10Z,13Z,15E, 19Z-docosahexaenoic acid) and 10S, 17S-diHDPA (10S,17S-dihydroxy-7Z,11E,13Z,15E,19Z-docosapentaenoic acid) were the most important mediators in predicting treatment responsiveness, with 15R-LXA4 (5S,6R,15R-trihydroxy-7E,9E,11Z,13E-eicosatetraenoic acid), 5S,12S-diHETE (5S,12S-dihydroxy-6E,8Z,10E,14Z-eicosatetraenoic acid), 4S, 14S-diHDHA (4S,14S-dihydroxy-5E,7Z,10Z, 12E, 16Z, 19Z-docosahexaenoic acid) and n-3 DPA-derived Maresin 1 (MaR1$_{n-3\ DPA}$) (7R,14S-dihydroxy-8E,10E,12Z,16Z,19Z-docosapentaenoic acid) also displaying a marked contribution, although to a lesser extent than the RvD4 and 10S, 17S-diHDPA (Fig. 1f). Having identified potential candidate biomarkers, we next built machine-learning models using the random forest methodology and concentrations of either RvD4, 10S, 17S-diHDPA, 15R-LXA4 and MaR1$_{n-3\ DPA}$ or RvD4, 10S, 17S-diHDPA, 15R-LXA4, 5S,12S-diHETE, 4S,14S-diHDHA and MaR1$_{n-3\ DPA}$. Using a second group of DMARD naive patients we then tested the ability of this model to assign patients to the correct outcome group (see Supplementary Tables 4 and 5 for patient clinical parameters and lipid mediator concentrations). Here we found that the combination of the top six mediators predicted treatment outcome in ~86% of the cases while the model build using the four mediators gave a prediction score of ~83% (Fig. 1g). We next validated the accuracy of these two models using mediator concentrations from a different group of DMARD naive patients. Results from these analyses demonstrated that the model built using the four mediators gave an AUC of 0.80, whereas the model built with the six mediators gave an AUC score of 0.79 (Fig. 1h). Of note, the AUC for these mediators were markedly better than those obtained using mediator concentrations from the n-3 DPA metabolome and disease scores (Fig. 1d).

**Increased lipid mediators in plasma from DMARD non-responders.** To gain insights into mechanisms determining the responsiveness of patients to DMARD treatment, we conducted lipid mediator pathway analysis to identify which pathways were differentially regulated between the two patient groups. This demonstrated that there was an upregulation of SPM biosynthetic pathways, including the DHA-derived RvD4 and the n-3 DPA-derived MaR1$_{n-3\ DPA}$ in DMARD non-responders.

These increases were coupled with an upregulation of pro-inflammatory eicosanoids, including the nociceptive mediators PGD$_2$ and PGE$_2$, in these patients when compared with DMARD responders (Fig. 2). To determine whether the differences in SPM expression were linked with a distinct transcriptional regulation of enzymes involved in SPM biosynthesis we assessed the transcript expression of ALOX enzymes in peripheral blood from these two patient groups. ALOX5, ALOX12, ALOX15 and ALOX15B transcript levels were similar between the DMARD responders and DMARD non-responders (Supplementary Fig. 7). These results suggest that regulation of SPM biosynthetic pathways may be via either the regulation of protein translation or post-translational modification of the enzymes to regulate their activity[22,23]. Thus, we next tested whether the activity of these enzymes was altered. For this purpose, we measured plasma levels of monohydroxylated fatty acids from all four fatty acid metabolomes to gain insights into their activity in the two patient groups. Assessment of plasma concentrations of 5-HETE, 5-HEPE, 7-HDPA and 7-HDHA, markers of ALOX5 activity, revealed a significant upregulation of 7-HDHA, 5-HEPE and 5-HETE in DMARD non-responders when compared with responders. Concentrations of markers for ALOX12 (14-HDPA and 14-HDHA) and ALOX15 (17-HDPA, 17-HDHA, 15-HEPE and 15-HETE) indicated an increase in activity for these enzymes in non-responders, given the increased levels of these molecules in plasma from these patients when compared with those found in responders (Supplementary Fig. 8). To further evaluate the origin of these proposed pathway markers we conducted chiral liquid chromatography-tandem mass spectrometry, which permits the separation the R and S isomers of a given hydroxylated fatty acid. Here we found that in plasma of both DMARD responders and DMARD non-responders the most abundant isomer for all monohydroxylated fatty acids tested was that carrying the alcohol in the S conformation (Supplementary Figs. 9–12 and Supplementary Table 6). Given that all four ALOX enzymes involved in SPM biosynthesis preferentially oxygenate fatty acids in the S conformation[24,25], these results indicate an increased ALOX activity in non-responders.

**Baseline lipid mediators are linked with distinct disease pathotypes.** Synovial molecular and histological features patients with RA can be classified into three categories lympho-myeloid, diffuse-myeloid and pauci-immune-fibroid. These pathotypes are associated with distinct disease evolution and responses to DMARD treatment[5]. Therefore, we next questioned whether immune-related features in each of these groups extended beyond the synovium into the systemic circulation. To address this question, we conducted lipid mediator profiling to assess whether peripheral blood lipid mediator concentrations in RA patients from each of these three groups were distinct prior to the initiation of DMARD therapy. Using PLS-DA we found that plasma lipid mediators were indeed characteristic for different disease pathotypes, i.e. distinct lipid mediator profiles clustered with each category (Fig. 3a). Assessment of the variable importance in projection (VIP) scores, which identify the contribution of each mediator in the observed separation between groups demonstrated an upregulation of pro-resolving mediators, including 15R-LXA4 and MCTR2 (13R-cysteinylglycinyl, 14S-hydroxy-4Z,7Z,9E,11E,13R,14S,16Z,19Z-docosahexaenoic acid), in peripheral blood from patients with the pauci-immune-fibroid pathotype. In plasma from these patients we also found an upregulation of pro-inflammatory and immunosuppressive mediators including PGD$_2$ and TxA$_2$, measured as its stable further metabolite TxB$_2$ (Fig. 3b).

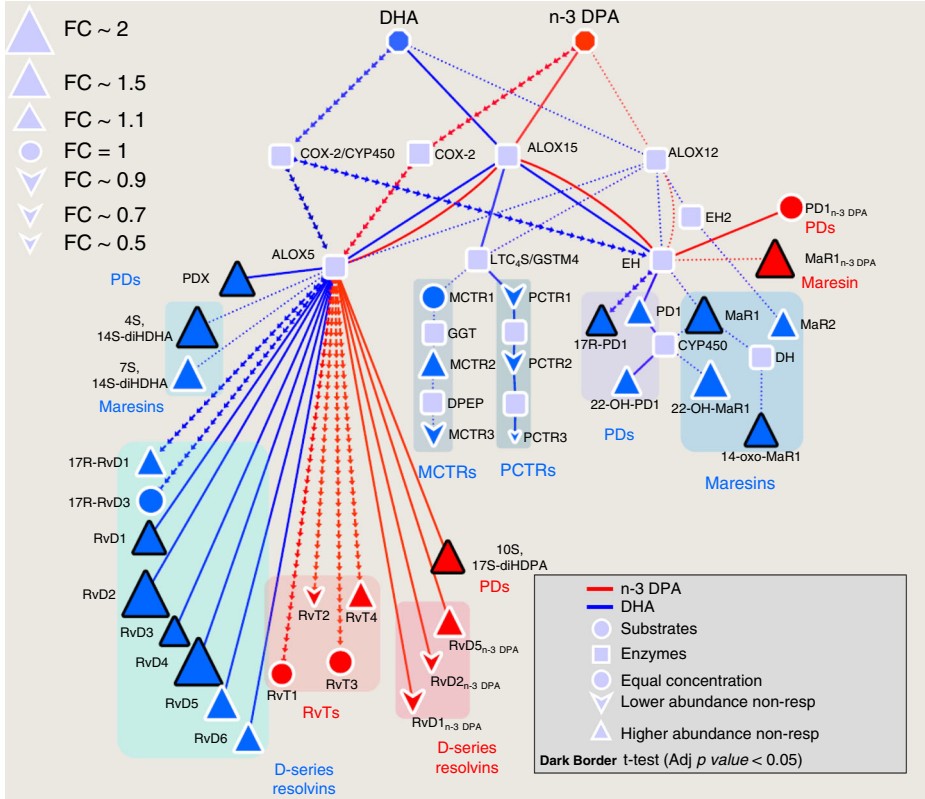

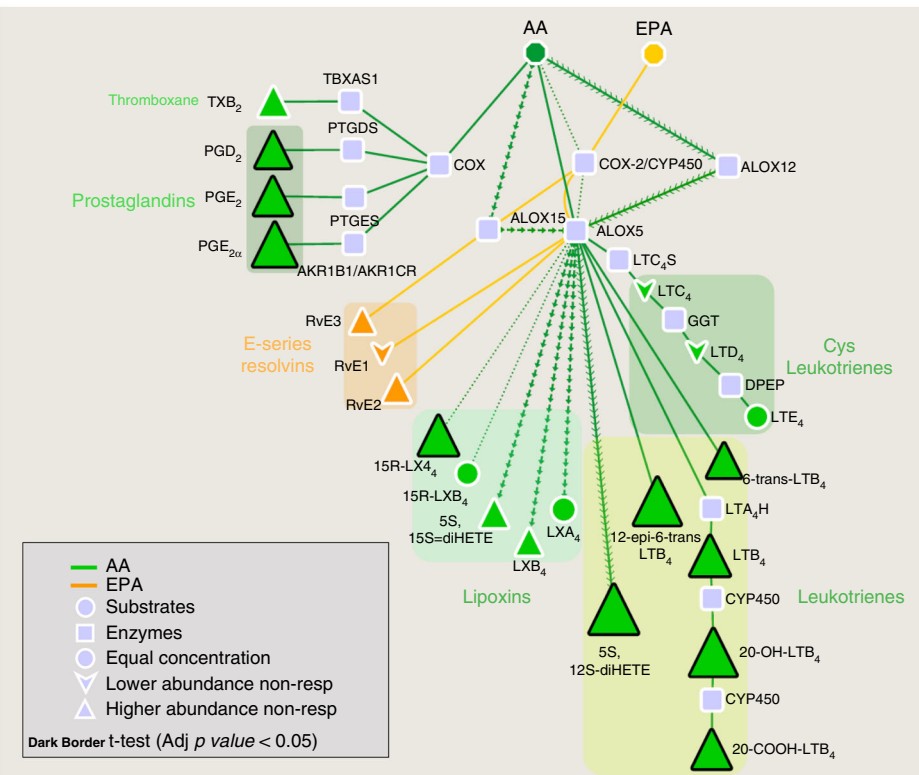

We next investigated the differential regulation of lipid mediator profiles between DMARD responders and non-responders for each of these three pathotypes. Here we found an increase in ALOX5 products from both the n-3 DPA and DHA metabolomes in non-responders with lympho-myeloid and those with a pauci-immune-fibroid pathotype when compared with responders with the respective pathotypes. These included

significant increases in the DHA-derived RvD4 and PDX. In these patients we also found a significant increase in n-3 DPA-derived $MaR1_{n-3\ DPA}$ (Fig. 3c). Assessment of mediators from the AA and EPA metabolomes demonstrated an increase in ALOX5 products in non-responders with lympho-myeloid and pauci-immune-fibroid pathotypes that reached statistical significance in patients with a pauci-immune-fibroid pathotype for the leukotriene (LT)

**Fig. 2 Upregulation of baseline peripheral blood lipid mediators in DMARD non-responders.** Peripheral blood was collected in patients DMARD responders (Resp) and DMARD non-responders (Non-Resp) prior to DMARD treatment initiation. Peripheral blood lipid mediator profiles were established in accordance with published criteria including matching retention time and MS/MS fragmentation spectra. Pathway analysis for the differential expression of mediators from the (top panel) DHA and n-3 DPA, and (bottom panel) EPA and AA bioactive metabolomes in Non-Resp when compared to Resp. Statistical differences between the normalised concentrations (expressed as the fold change) of the lipid mediators from the Non-Resp and Resp groups were determined using a two-sided $t$ test followed by a multiple comparison correction using Benjamini–Hochberg procedure. Up- or downregulated mediators are denoted with using upward and downward facing triangles, respectively, and on changes of the node's size. Bolded mediators represent statistical differences between the two groups when adjusted $p$ value <0.05. Results are representative of $n = 66$ for Resp and $n = 44$ Non-Resp. Source data are provided as a Source data file.

$B_4$ pathway, including $LTB_4$ and its further metabolite 20-COOH-$LTB_4$. In these patients, we also found a statistically significant increase in concentrations of the pro-inflammatory and nociceptive mediator $PGE_2$ (Supplementary Fig. 13). These results demonstrate that differences in peripheral blood lipid mediator profiles between DMARD responders and non-responders are common to different RA pathotypes.

Having found that lipid mediator concentrations were different between these patient groups, we next assessed whether combining disease pathotypes with the biomarkers identified above would further enhance the predictiveness of our machine-learning model. Results from this analysis demonstrate a marked increase in the predictiveness of RvD4, 10S, 17S-diHDPA, 15R-$LXA_4$ and $MaR1_{n-3\ DPA}$ at identifying responders, when separate machine-learning models were built for each of the RA pathotypes, with the ability of the model to correctly classify responders increasing to ~89% (Fig. 3d).

**Differences in SPM concentrations are maintained after treatment initiation.** Having observed a differential regulation in peripheral blood SPM concentrations between DMARD responders and DMARD non-responders prior to treatment initiation, we next investigated whether differences in peripheral blood lipid mediator concentrations were also present in patients 6 months after treatment initiation. OPLS-DA analysis demonstrated that plasma lipid mediator profiles from DMARD responders 6 months after the initiation of treatment were distinct from those of DMARD non-responders, as demonstrated by a separation between the two patient clusters (Fig. 4a, Supplementary Table 7). Assessment of VIP scores identified 22 mediators and SPM pathway markers that were differentially expressed between the two patient groups (Fig. 4b). Amongst these mediators we found SPM that are involved in coordinating the host response during ongoing inflammation such as PCTR2 (16-cysteinylglycinyl, 17S-hydroxy-4Z,7Z,11,13,15E,19Z-docosahexaenoic acid), RvD2 (7S,16R,17S-trihydroxy-4Z,8E,10Z,12E,14E,19Z-docosahexaenoic acid) and RvD3 (4S,11R,17S-trihydroxy-5Z,7E,9E,13Z,15E,19Z-docosahexaenoic acid)[16,22,26] as well as mediators linked with pain modulation e.g. RvE2[11] (Fig. 4b).

In order to gain further insights into the mediator pathways that were differentially regulated between these patient groups, we interrogated the biosynthetic pathways for each of the essential fatty acid metabolomes. This analysis demonstrated significant increases in concentrations of select ALOX5 and ALOX15-derived mediators from the DHA metabolome that included RvD1 and 17R-PD1 (10R,17R-dihydroxy-7Z,11E,13E,15Z,19Z-docosahexaenoic acid) in plasma of non-responders when compared to responders (Fig. 4c). Pathway analysis of EPA and AA-derived lipid mediator concentrations demonstrated that while ALOX5 derived products of EPA were also reduced, AA-derived ALOX5 products, including those of the potent leucocyte chemoattractant $LTB_4$ and the ionotropic cysteinyl leukotrienes[27], were markedly increased in plasma from non-responders when compared with responders (Fig. 4d).

Having observed significant changes in SPM concentrations, we next investigated whether the activity of ALOX enzymes and the conversion of DHA and n-3 DPA was altered in peripheral blood cells from the two patient groups. For this purpose, we measured plasma levels of monohydroxylated fatty acids from the DHA and n-3 DPA metabolomes to gain insights into both enzyme activity and substrate conversion. Plasma concentrations of the ALOX5 products 7-HDPA and 7-HDHA were either similar (7-HDPA) between the two patient groups, or upregulated (7-HDHA) in DMARD non-responders. Concentrations of the ALOX12 (14-HDPA and 14-HDHA) and ALOX15 (17-HDHA and 17-HDPA) products were increased in non-responders (Supplementary Fig. 14). Of note, as observed in baseline plasma, chiral analysis of monohydroxylated fatty acids demonstrated that the predominant isomer for these products was the S-isomer (Supplementary Table 8). These findings indicate that the observed reduction in plasma DHA and n-3 DPA-derived SPM in DMARD non-responders was not due to a decrease in ALOX activity and/or substrate availability/conversion in peripheral blood cells from these patients. Together these observations demonstrate that 6 months after treatment initiation plasma SPM concentrations in DMARD responders were higher than those measured in DMARD non-responders. Given that enzyme activity was elevated in non-responders when compared with responders, this suggests that uncoupling of the SPM biosynthetic pathways may be responsible for the reductions in plasma SPM concentrations.

## Discussion

The present findings uncover a previously unappreciated role for SPM as predictive biomarkers to DMARD responsiveness in RA. Assessment of baseline plasma SPM demonstrated that the concentrations of select mediators were predictive of DMARD treatment responsiveness, identifying novel functional biomarkers. Differences in plasma SPM concentrations were found to persist to 6 months after the initiation of DMARD treatment in non-responders when compared with responders.

Mounting evidence implicates a role for altered resolution mechanisms in the onset and propagation of RA. Increasing synovial RvE2 concentrations were found to correlate with decreased joint pain, whereas plasma SPM concentrations were negatively related to erythrocyte sedimentation rate[11]. In experimental systems 17R-RvD1 (7S,8R,17R-trihydroxy-4Z,9E,11E,13Z,15E,19Z-docosahexaenoic acid; also referred to as aspirin-triggered-RvD1) attenuates arthritis severity, cachexia, hind-paw oedema, and paw leucocyte infiltration, shortening the remission interval[19]. RvD3 concentrations are reduced in inflamed joints from mice with delayed-resolving arthritis when compared with self-resolving inflammatory arthritis. Administration of this mediator to arthritic mice reduced joint leucocyte trafficking, joint eicosanoid concentrations, and joint inflammation[16]. RvD1 (7S,8R,17S-trihydroxy-4Z,9E,11E,13Z,15E,19Z-docosahexaenoic acid) and its precursor 17-HDHA were also found to display anti-hyperalgesic properties[28]. In the present study, we

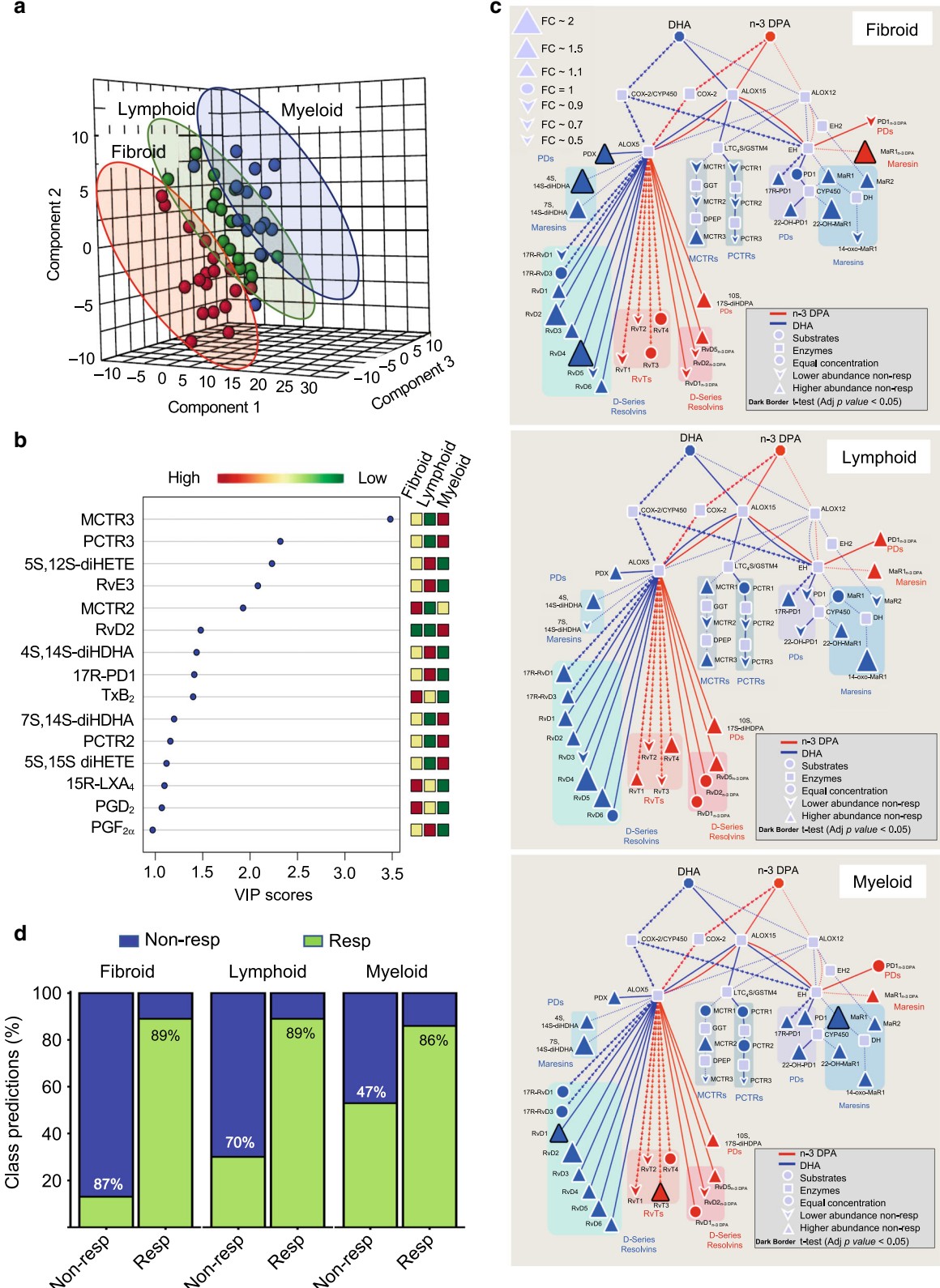

found that peripheral blood concentrations of both SPM and inflammatory eicosanoids were increased in DMARD non-responders at baseline (Figs. 1 and 2). These changes were independent of differences in overall circulating platelet and leucocyte numbers, suggesting that they may reflect a differential activation status in peripheral blood leucocytes as previously reported for other leucocyte subsets[13]. Furthermore, the

concentrations of select SPM were higher in patients that were non-responsive to DMARDs when compared with those that were responsive 6 months after treatment initiation (Fig. 4). Given that one of the key biological actions of SPM is to counter-regulate eicosanoid production[6,10,15,16,19,22,29], these findings suggest that SPM activity in DMARD non-responders may be compromised. This observation is in line with findings made in

**Fig. 3 Combining disease pathotypes and select SPM concentrations enhances model predictiveness.** Plasma was collected from RA patients prior to the initiation of treatment with DMARD and lipid mediator concentrations established using LC-MS/MS-based lipid mediator profiling. **a, b** PLS-DA analysis of peripheral blood lipid mediator concentrations for lympho-myeloid (lymphoid), diffuse-myeloid (myeloid) and pauci-immune-fibroid (fibroid) pathotypes. **a** 3-dimensional score plot. **b** Variable importance in projection (VIP) scores of 15 lipid mediators with the greatest differences in concentrations between the three groups. Results are representative of $n = 18$ for Fibroid $n = 17$ for myeloid and $n = 19$ for lymphoid. **c** Pathway analysis for the differentially expressed mediators from the DHA and n-3 DPA bioactive metabolomes in DMARD non-responders (Non-Resp) when compared to DMARD responders (Resp) for each pathotype. Statistical differences between the normalised concentrations (expressed as the fold change) of the lipid mediators from the Non-Resp and Resp groups were determined using two-sided $t$ test followed by a multiple comparison correction using Benjamini–Hochberg procedure. Up- or downregulated mediators are denoted with using upward and downward facing triangles, respectively, and on changes of the node's size. Bolded mediators represent statistical differences between the two groups when adjusted $p$ value <0.05. Results are representative of $n = 18$ for fibroid Resp, $n = 15$ for fibroid Non-Resp, $n = 19$ for lymphoid Resp, $n = 10$ for lymphoid Non-Resp, $n = 22$ for myeloid Resp, $n = 15$ for myeloid Non-Resp. **d** Classification accuracies for each class (sensitivity and specificity) of the RvD4, 10S,17S-diHDPA, 15R-LXA$_4$ and MaR1$_{n-3\ DPA}$ model created using the specific dataset for the different pathotypes (fibroid, lymphoid and myeloid). Green indicates the samples that were predicted as Resp while blue indicates predicted Non-Resp. Percentages indicate true positives (Resp class) and true negatives (Non-Resp class). All the models were created using the random forest methodology ("randomForest" package from R). Source data are provided as a Source data file.

diabetic patients where the signalling downstream of the RvE1 receptor, chemerin chemokine-like receptor 1, was found to be altered reducing the ability of RvE1 to regulate peripheral blood leucocyte responses from these patients[30].

It is widely believed that a precision medicines approach is likely to be more effective in the treatment of patients with chronic inflammatory disorders, including those with RA, than the current process where patients are treated in a prescriptive manner using a *one size fits all* approach[31]. Unfortunately, the lack of robust biomarkers to determine treatment responsiveness in many chronic inflammatory conditions, including RA, has hindered the development of this approach in clinical settings[32]. Results from the present study demonstrate that plasma lipid mediator concentrations prior to the initiation of treatment are different in patients that respond to DMARDs when compared with those that do not. Using machine-learning methodologies, we found that the concentrations of a select group of mediators were predictive of treatment responsiveness in two RA cohorts, with prediction accuracy of up to ~89% (Figs. 1 and 3). Furthermore, plasma SPM concentrations prior to DMARD treatment initiation were also diagnostic of distinct joint disease pathotypes (Fig. 3). Importantly, while patients enrolled in this study were DMARD naive, most of the patients in both cohorts were on a wide range of other medications for a number of co-morbidities, although there were no significant differences in any of these parameters between the two patient groups (see Supplementary Tables 1 and 4). Therefore, the identification of a specific lipid mediator signature that is predictive of DMARD responsiveness suggests that changes in these lipid mediators are specific for this group of therapeutics. In addition, since SPM regulate host innate and adaptive immune responses and their production is reflective of leucocyte activation status[13–15], the present findings indicate that peripheral blood SPM concentrations are potential functional biomarkers for both patient stratification and predicting treatment responsiveness to DMARDs.

The biosynthesis of SPM involves the stereoselective conversion of essential fatty acids by distinct enzymes, with the successful formation of the bioactive product relying on the expression and activity of the enzyme as well as their appropriate subcellular localization. In this context studies investigating mechanisms regulating SPM biosynthesis demonstrated that activation of MAPK leads to enzyme phosphorylation at serine 271 that is subsequently translocated to the nuclear membrane where it couples with phospholipase A$_2$ and Leukotriene A$_4$ Hydrolase to produce leukotrienes[27,33]. On the other hand, in the absence of phosphorylation the enzyme is retained in the cytosol where it couples with ALOX15 to produce SPM[33]. During ongoing inflammation, for example in atherosclerosis, an increase

in the expression of phosphorylated ALOX5 and a decrease in the RvD1 to LTB$_4$ ratio was observed[33]. These findings indicate that in addition to expression of the enzyme, post-translational modifications of the protein are central in determining the product profile of the enzyme, that is, whether the enzymes produce SPM or pro-inflammatory eicosanoids. In the present study, we found that concentrations for most SPM identified in the plasma of DMARD non-responders were either similar to those found in responders or reduced. This observation was coupled with an increase in ALOX activity (Fig. 4, Supplementary Fig. 14 and Supplementary Table 8), suggesting that the SPM biosynthetic pathways become uncoupled post DMARD treatment initiation in non-responders.

In summation, the present study identifies novel functional biomarkers, including RvD4, 10S, 17S-diHDPA, 15R-LXA$_4$ and MaR1$_{n-3\ DPA}$, that predict both treatment response to DMARDs as well as joint disease pathotype. Thus, these biomarkers may be clinically useful in identifying patients who are unlikely to respond to conventional DMARD therapy and would benefit from being fast-tracked to the next level of RA therapeutics. This would in turn help minimise or even prevent further structural damage to the joints together with disease progression and disability, thereby improving quality of life.

## Methods
**Materials**. Liquid chromatography (LC)-grade solvents were purchased from Fisher Scientific (Pittsburgh, PA, USA); Poroshell 120 EC-C18 column (100 mm × 4.6 mm × 2.7 μm) was obtained from Agilent (Cheshire, UK); C18 SPE columns were from Biotage (Uppsala, SE); synthetic standards for LC-tandem mass spectrometry (MS-MS) quantitation and deuterated (d) internal standards (d$_8$-5S-HETE (Cat no: CAY334230); d$_5$-RvD2 (Cat no: CAY11184); d$_5$-LXA$_4$ (Cat no: CAY10007737); d$_4$-PGE$_2$ (Cat no: CAY314010); d$_4$-LTB$_4$ (Cat no: CAY320110); d$_5$-LTC$_4$ (Cat no: CAY10006198); d$_5$-LTD$_4$ (Cat no: CAY10006199); d$_5$-LTE$_4$ (Cat no: CAY10007858)) and synthetic lipid mediator standards (RvD1, CAY10012554; 17R-RvD1 (Cat no: CAY13060); RvD2 (Cat no: CAY10007279); RvD3 (Cat no: CAY13834); 17R-RvD3 (Cat no: CAY9002880); RvD4 (Cat no: CAY13835); RvD5 (Cat no: CAY10007280); MaR1 (Cat no: CAY10878); MaR2 (Cat no: CAY16369); MCTR1 (Cat no: CAY17007); MCTR2 (Cat no: CAY17008); MCTR3 (Cat no: CAY19067); PDX (Cat no: CAY10008128); PCTR1 (Cat no: CAY19064); PCTR2 (Cat no: CAY19065); PCTR3 (Cat no: CAY19066); 4-HDHA (Cat no: CAY33200); 7-HDHA (Cat no: CAY33300); 14-HDHA (Cat no: CAY33550); 17-HDHA (Cat no: CAY33650); RvE1 (Cat no: CAY10007848); 5-HEPE (Cat no: CAY32210); 12-HEPE (Cat no: CAY32540); 15-HEPE (Cat no: CAY32700); 18-HEPE (Cat no: CAY32840); RvD5$_{n-3\ DPA}$, CAY10546; LXA$_4$ (Cat no: CAY90410); 15-epi-LXA$_4$ (Cat no: CAY90415); LXB$_4$ (Cat no: CAY90420); 5S,15S-diHETE (Cat no: CAY35280); PGD$_2$ (Cat no: CAY12010); PGE$_2$ (Cat no: CAY14010); PGF$_{2\alpha}$ (Cat no: CAY16010); TXB$_2$ (Cat no: CAY19030); LTB$_4$ (Cat no: CAY20110); 6-trans-LTB$_4$ (Cat no: CAY35250); 6-trans,12-epi-LTB$_4$ (Cat no: CAY35265); 20-OH-LTB$_4$ (Cat no: CAY20190); 20-COOH-LTB$_4$ (Cat no: CAY20180); LTC$_4$ (Cat no: CAY20210); LTD$_4$ (Cat no: CAY20310); LTE$_4$ (Cat no: CAY20410); 5-HETE (Cat no: CAY34210); 12-HETE (Cat no: CAY34550); 15-HETE (Cat no: CAY34700)) were purchased from Cambridge Bioscience (Cambridge, UK) or provided by Charles N. Serhan (Harvard Medical School, Boston, Massachusetts, USA;

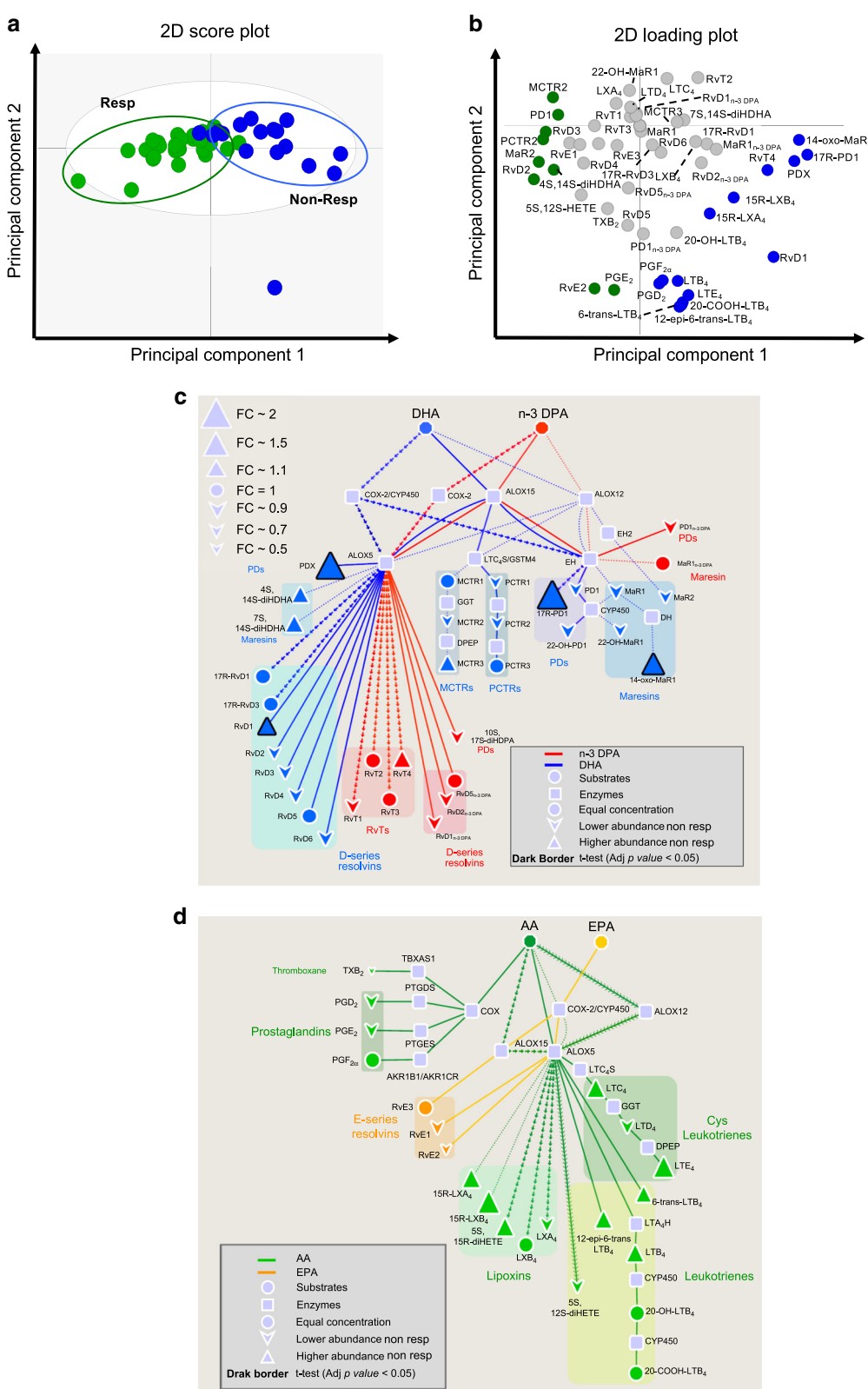

supported by NIH-funded P01GM095467); Dulbecco's phosphate-buffered saline (DPBS, without calcium and magnesium, Sigma (Cat no: D8537)).

**Pathobiology of Early Arthritis Cohort**. Plasma samples were taken at baseline and 6 months from 112 and 44 patients, respectively. These were obtained from the Pathobiology of Early Arthritis Cohort (PEAC). The PEAC cohort study was approved by the King's College Hospital Research Ethics Committee (REC 05/Q0703/198). Patients provided informed consent. Peripheral blood samples and

synovial tissue were obtained from patients recruited at Barts Health NHS Trust into the Pathobiology of Early Arthritis Cohort (PEAC, http://www.peac-mrc.mds.qmul.ac.uk) undergoing ultrasound (US)-guided synovial biopsy of the most inflamed joint (knee, wrist or small joints of hands or feet)[5]. All patients were DMARDs and steroid-naive, had symptoms duration <12 months and fulfilled the ACR/EULAR 2010 classification criteria for RA. RA individuals were categorised into three pathotypes based on histological classification of synovial tissue: lympho-myeloid, diffuse-myeloid and pauci-immune-fibroid (for more details see ref. [5]).

**Fig. 4 Decreased SPM levels in DMARD non-responders 6 months after treatment initiation.** Peripheral blood was collected in patients that displayed reduced joint disease (DMARD responders, Resp) and those that did not (DMARD non-responders; Non-Resp) 6 months after treatment initiation. Peripheral blood lipid mediator profiles were established using LC-MS/MS-based lipid mediator profiling. **a, b** OPLS-DA of lipid mediator profiles from Resp and Non-Resp. **a** Score plot, **b** Represents the loading plot with mediators displaying VIP score >1 highlighted in green or blue and correspond with either Resp or Non-Resp, respectively. **c, d** Pathway analysis for the differential expression of mediators from the **c** DHA and n-3 DPA and **d** EPA and AA bioactive metabolomes in Non-Resp when compared to Resp. Statistical differences between the normalised concentrations (expressed as the fold change) of the lipid mediators from the Non-Resp and Resp groups were determined using a two-sided $t$ test followed by multiple comparison correction using Benjamini–Hochberg procedure. Up- or downregulated mediators are denoted with using upward and downward facing triangles, respectively, and on changes of the node's size. Bolded mediators represent statistical differences between the two groups when adjusted $p$ value <0.05. Results are representative of $n = 27$ for Resp and $n = 17$ for Non-Resp. Source data are provided as a Source data file.

Patients were treated with DMARDs. Response status after 6 months of mixed DMARD therapy was determined by EULAR response criteria based on DAS28-ESR.

**Targeted lipid mediator profiling.** Plasma was obtained from peripheral blood following centrifugation at $1500 \times g$ for 10 min at room temperature. All samples were extracted using solid-phase extraction columns as in refs. [20,34]. A step-by-step description of the extraction, analysis and quantitation procedures are detailed in the following protocol found in Protocol Exchange[35]. Prior to sample extraction, deuterated internal standards, representing each region in the chromatographic analysis (500 pg each) were added to facilitate quantification. Samples were kept at $-20\,°C$ for a minimum of 45 min to allow protein precipitation. Supernatants were subjected to solid-phase extraction, methyl formate fraction collected, brought to dryness and suspended in phase (methanol/water, 1:1, vol/vol) for injection on a Shimadzu LC-20AD HPLC and a Shimadzu SIL-20AC autoinjector, paired with a QTrap 5500 or QTrap 6500+ (Sciex). An Agilent Poroshell 120 EC-C18 column (100 mm × 4.6 mm × 2.7 μm) was kept at $50\,°C$ and mediators eluted using a mobile phase consisting of methanol/water/acetic acid of 20:80:0.01 (vol/vol/vol) that was ramped to 50:50:0.01 (vol/vol/vol) over 0.5 min and then to 80:20:0.01 (vol/vol/vol) from 2 min to 11 min, maintained till 14.5 min and then rapidly ramped to 98:2:0.01 (vol/vol/vol) for the next 0.1 min. This was subsequently maintained at 98:2:0.01 (vol/vol/vol) for 5.4 min, and the flow rate was maintained at 0.5 ml/min. QTrap 5500 or QTrap 6500+ were operated using a multiple reaction monitoring method as in refs. [20,34]. Supplementary Tables 9 and 10 report instrument source parameters and Supplementary Tables 11–12 report coefficient of variation for sMRM transitions employed in the quantitation of lipid mediators. Each lipid mediator was identified using established criteria, these included: (1) presence of a peak with a minimum area of 2000 counts, (2) matching retention time to synthetic or authentic standards with maximum drift between the expected retention time and the observed retention time of 0.05 s, (3) ≥4 data points, and (4) matching of at least 6 diagnostic ions to that of reference standard, with a minimum of one backbone fragment being identified in reperesntative samples[13,20,34]. Calibration curves were obtained for each mediator using lipid mediator mixtures at 0.78, 1.56, 3.12, 6.25, 12.5, 25, 50, 100 and 200 pg that gave linear calibration curves with an $r^2$ values of 0.98–0.99. Signal-to-noise ratio was calculated using the Signal-to-Noise script from Analyst (version 1.6.3, Sciex). Here the application intensity value for the region denoted as the signal/peak of interest by the intensity value for the highest peak in the region denoted as the noise.

**Chiral LC-MS/MS analysis.** Step-by-step description of the extraction, analysis and quantitation procedures are detailed in the following protocol found in Protocol Exchange[35]. Briefly, a Chiralpak AD-RH column (150 mm × 2.1 mm × 5 μm) was used with isocratic methanol/water/acetic acid 95:5:0.01 (v/v/v) at 0.15 ml/min. To monitor isobaric monohydroxy fatty acid levels, a multiple reaction monitoring (MRM) method was developed using signature ion fragments for each molecule as in ref. [22].

**Description of data used for model building.** The data used for the machine-learning models and network analyses consisted of the lipid mediator profiles of patients with RA who responded ($n = 30$) or did not ($n = 24$) to the treatment with DMARDs for the first PEAC-derived patient cohort. The lipid mediator profile included DHA-derived resolvins (RvD1, RvD2, RvD3, RvD4, RvD5, RvD6, 17-RvD1 and 17-RvD3), protectins (PD1, 17-PD1, 10S,17S-diHDHA, also known as PDX, and 22-OH-PD1), PCTRs (PCTR1, PCTR2 and PCTR3), maresins (MaR1, MaR2, 7S,14S-diHDHA, 4S,14S-diHDHA, 14-oxo-MaR1 and 22-OH-MaR1), MCTRs (MCTR1, MCTR2 and MCTR3), 13-series resolvins (RvT1, RvT2, RvT3 and RvT4), n-3 DPA-derived resolvins (RvD1$_{n-3\ DPA}$, RvD2$_{n-3\ DPA}$ and RvD5$_{n-3\ DPA}$), n-3-DPA-derived protectins (PD1$_{n-3\ DPA}$ and 10S, 17S-diHDPA), n-3 DPA-derived maresins (MaR1$_{n-3\ DPA}$), E-series resolvins (RvE1, RvE2 and RvE3), leukotrienes (LXA$_4$, LXB$_4$, 5S,15S-diHETE, 20-OH-LTB$_4$, 20-COOH-LTB$_4$, 6-trans-LTB$_4$ and 12-epi-6-trans-LTB$_4$), cysteinyl leukotrienes (LTC$_4$, LTD$_4$ and LTE$_4$), prostaglandins (PGD$_2$, PGE$_2$ and PGF$_{2\alpha}$) and thromboxane (TXB$_2$). A Clinical Score model was obtained using the following clinical parameters: disease duration, erythrocyte sedimentation rate (ESR), rheumatoid factor (RF titre), tiredness visual

analogue scale (VAS), pain VAS, patient global health VAS, physician global assessment VAS, swollen joints number, disease activity score-28 (DAS28) and 12 max US Synovial Thickness and US Power Doppler scores. A second patient cohort of 58 patients (36 responders and 22 non-responders) was obtained from the PEAC study and was used as the test dataset for the lipid mediator profiling and Clinical Score models, and also as the training dataset for improved models based on specific biomarkers. Age, sex and clinical parameters not mentioned before were not considered for this first approximation of creating a model able to classify the response of RA patients to DMARD treatment.

**Model building.** Data were pre-processed and analysed using R Software (v3.5.1; https://www.r-project.org/) and RStudio environment (v1.1.456; https://www.rstudio.com/).

From the exploratory analysis, two samples were removed for showing outlier concentrations of TXB$_2$, which likely reflected coagulation during sample collection and an additional sample was removed due to lack of clinical records. Although no normalization was required since all the lipid mediator concentrations were calculated based on the same amount of standard, the concentrations were scaled by subtracting the mean and dividing by the standard deviation of each feature.

Two supervised machine-learning methodologies were used to create the classifier models: support vector machine (SVM)[36] and randomForest[37]. SVM separates groups by organizing the samples in two spaces divided by a hyperplane in a way that the distances between the samples in the same group are not too wide and the distance between the groups is as large as possible[38]. A nonlinear kernels radial basis function SVM was used. In order to identify the best model, we created models testing different times of the resampling and different number of ensembles (fusion of the individual classifiers created during the bootstrapping step) with 70 bootstrap iterations and 70 individual classifiers in each ensemble that gave stable models for all groups tested. Furthermore, we also used the inbuilt automatic optimization step that includes minimization of the bootstrapping error[36] in the R Package "classyfire" (https://cran.r-project.org/src/contrib/Archive/classyfire/), to improve and validate the models (see Supplementary Fig. 6 for representative outcomes).

RandomForest operates by getting the consensus of weak decision tree classifiers[39]. The decision trees are created using the features as vertices and classes as leaves; each tree is designed using a different set of randomly chosen features[40]. In the present studies using the R package "randomForest" (https://cran.r-project.org/package=randomForest), which uses bootstrapping as the test method, we first used a small loop to test the different *mtry* values, the number of variables randomly sampled as candidates at each split. Then using the *mtry* value that gave the best classification performance for each model we tested a number of *ntree* values, the number of decision trees that are created before creating the consensus classifier tree, to obtain the most stable tree.

Here we found that an *ntree* of 10,000 gave us the most stable models for all the lipid mediator groups and for the majority of the variables tested. Increasing the number of *ntrees* beyond this value did not markedly improve the outcomes (see Supplementary Fig. 6 for representative outcomes).

**Model evaluation.** Receiver operating characteristic curves (ROC curves) were built to evaluate the prediction accuracy of the models when predicting between DMARDs responder and DMARDs non-responder in a test dataset. ROC curves are created by plotting the true positive rate against the false-positive rate, showing the sensitivity and specificity of the model, when the discrimination threshold changes. The area under the curve (AUC) is calculated as the prediction performance of the models. ROC curves were created using the R package "pROC" (https://cran.r-project.org/web/packages/pROC/index.html). AUC values close to 1 (AUC > 0.8) refer to good classifier models.

Alongside ROC curves, other statistics such as the percentage of correctly classified samples (% accuracy score), specificity and sensitivity were also calculated.

**Feature selection and model improvement.** As random forests showed the best validation scores during the testing step, the model improvement was based on the RandomForest methodology. The lipid mediators were separated in groups based

on their precursors (DHA, n-3 DPA, EPA and AA) or the distinct clusters of mediators. Different models were created using only the most relevant lipid mediators and the "importance" function of 'randomForest' package. This function organizes the model's features by relevance based on the model's decreased mean accuracy when the specific feature is not present. The % accuracy score and AUC (ROC) were calculated for all the models and, according to the results, the best models and the most relevant biomarkers for the classification of the DMARD-responder and DMARD non-responder patients were selected.

**Pathotypes analyses**. All the data (training and test cohorts) was separated based on the specific pathotype shown for the patients: pauci-immune/fibroid ($n = 28$), lympho-myeloid ($n = 27$) and diffuse-myeloid ($n = 31$). This was made with the purpose of seeking better classification models and seeing if specific lipid mediators were responsible for the different manifestation of the disease. The models were build using RandomForest and different statistic scores were calculated for the validation of each model.

**Network analyses**. Statistical differences between the normalised concentrations (expressed as the fold change) of the lipid mediators from the DMARD non-responder and DMARD-responder groups were determined using a two-sided $t$ test followed by a multiple comparison correction using Benjamini–Hochberg procedure. Based on these differences, lipid mediator bio-synthesis networks were constructed using Cytoscape v3.7.1. The different path-ways were illustrated using different colours and line shapes, while up- or downregulated mediators were denoted with using upward and downward facing triangles, respectively, and on changes of the node's size. Bolded mediators represent statistical differences between the two groups. The comparison between DMARD non-responders and DMARD responders were made with pre and post-treatment data.

**Enzyme transcript expression**. RNA was extracted from whole blood samples in RNALater solution using the Ambion Ribo-Pure Blood kit (ThermoFisher Scientific). Total RNA-sequencing (RNA-seq) was performed on an Illumina HiSeq2500 platform. Raw data were quality-controlled using FastQC, trimmed or removed with Cutadapt. Transcript abundance was derived from paired sample FASTQ files over GENCODE v24/GRCh38 transcripts using Kallisto v0.43.0. Normalization of the raw data and differential gene expression analysis between DMARD-responder and DMARD non-responder were performed using the quasi-likelihood method of the Bioconductor R package "edgeR" (https://bioconductor.org/packages/release/bioc/html/edgeR.html). Results are expressed as the log counts per million of each gene.

**Statistical analysis**. We performed all statistical analyses and data derivation using R v3.5.1[41], MetaboAnalyst v4.0[21], Prism v8 and Microsoft Excel Professional Plus 2016. Results presented in the figures are expressed as means and those displayed in the tables are displayed as mean ± sem.

Differences between groups were determined using two-sided $t$ test (normalized data) or Mann–Whitney test (2 groups). Sample sizes for each experiment were determined on the variability observed in prior experiments. Partial least squares-discrimination analysis (PLS-DA) and orthagonal partial least squares-discrimination analysis (OPLS-DA) were performed using MetaboAnalyst v4.0[21] or SIMCA v14.1 software (Umetrics, Umea, Sweden) after mean centring and unit variance scaling of lipid mediator concentrations. PLS-DA is based on a linear multivariate model that identifies variables that contribute to class separation of observations (e.g. treatment response) on the basis of their variables (lipid mediator concentrations). During classification, observations were projected onto their respective class model. The score plot illustrates the systematic clusters among the observations (closer plots presenting higher similarity in the data matrix).

**Reporting summary**. Further information on research design is available in the Nature Research Reporting Summary linked to this article.

## Data availability
All lipid mediator profiling data generated during and/or analysed in the current study are available in the following public Github repository (Section Data https://github.com/eagomezc/2019_Machine_Learning_DMARD_in_RA_patients/tree/master/a_Data) or from the corresponding author upon reasonable request. RNA-Seq data have been deposited in ArrayExpress under accession code E-MTAB-6141. A step-by-step description of the protocol employed for the identification and quantitation of lipid mediators has been published in Protocol Exchange (https://doi.org/10.21203/rs.3.pex-1147/v1). Source data are provided with this paper.

## Code availability
Codes employed for machine-learning analysis in this study are deposited in Github repository (section R Scripts): https://github.com/eagomezc/2019_Machine_Learning_DMARD_in_RA_patients/tree/master/b_R_Scripts.

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

## Acknowledgements

The authors would like to thank all the patients who participated in this study as well as Ms. Kimberly Pistorius and Dr Lucy Ly (Lipid Mediator Unit, QMUL) for technical assistance. This work was supported by funding from the European Research Council (ERC) under the European Union's Horizon 2020 research and innovation programme (grant no. 677542) and the Barts Charity (grant no. MGU0343) to J.D. J.D. is also supported by a Sir Henry Dale Fellowship jointly funded by the Wellcome Trust and the Royal Society (grant 107613/Z/15/Z). The Pathobiology of Early Arthritis Cohort (PEAC) was supported by funding from the UK Medical Research Council (grant code G0800648). Core work associated with this project was supported by grants from Arthritis Research UK (Experimental Arthritis Treatment Centre, grant code 20022) and Genentech.

## Author contributions

J.D. and C.P. designed the experiments and conceived the overall research plan; E.A.G., P.R.S., R.A.C., R.H. and M.L. conducted the experiments and/or analysed results; all authors contributed to manuscript preparation; C.P., C.B. and J.D. contributed to supervision of the work.

## Competing interests

J.D. declares being a scientific founder and director of Resolomics Limited. The other authors declare no competing interests.
