## [Peer Review File · Nature Communications]

Reviewers' comments:

Reviewer #1 (Remarks to the Author):

The authors have dropped the murine data from the manuscript and this corrects one of the major problems with the manuscript.

For one used to looking at bar graphs or curves when discussing changes in concentration of various reactants in patients before and after treatment it is a little difficult to follow the graphs as noted in Figs. 2 and 4. Perhaps a principal component analysis could be done comparing the four different groups (non-responders and responders at baseline, non-responders and responders after treatment). The point the authors make is valid and interesting (really the point of the entire work) but the changes are difficult to make out with the graphics and analyses used.

Finally, this reviewer has a minor quibble with the sentence added in page 3 lines 54-56. The point here was that CD39 levels on Tregs is a biomarker that predicts MTX responses and CD39 is critical for the formation of adenosine in the extracellular space (one of the mechanisms described in the cited review).

Reviewer #2 (Remarks to the Author):

The authors have been very responsive to the reviewer's comments and provided new data and analysis. Concerns about experiments that were aimed at defining a mechanism for MTX regulation of 15-LOX or SPM formation have been addressed by not including the data. Even though this eliminates a major weakness of the central hypothesis it also significantly lowers the potential impact of this study. The study is now limited to an elegant but descriptive lipidomic analysis but provides no logical mechanism to explain the coincident changes in plasma levels of selected SPM, a conclusion that is based on machine learning analysis. The changes in plasma SPM may have multiple and indirect causes, which is a significant limitation of the study.

Reviewer #3 (Remarks to the Author):

The actual version of the manuscript is improved but needs more clarifications and ameliorations. I have few comments:

1/ If possible to specify that you used the linear SVM.

2/ Dimension of the input dataset: better to include Number of individuals x Number of variables

3/ RandomForest: when hyper tuning the parameters, you fixed mtree=10000 because it gave you the best performing models when running all the experiments. What about the number of variables, did you also use the same number for training all the models?

4/ Table 1: Could you please specify the formula of MCC you used? How exactly did you compute it using the mltools? Was it based on the values of TP, FP, TN and FN? If yes, it is possible to compute them manually and you will obtain different results.

The MCC values obtained from running the models on Cohort 1 are different between the first and the revised version, but we have the same values for the other evaluation metrics, why? As far as I understood, Cohort 1 didn't change, right?

How is it possible to have high mean sensitivity and specificity values and quite low AUC value, as for the DHA model?

5/ Most relevant biomarkers: You selected the best metabolites DHA and EPA based mainly on the values of accuracy? Why this metric is more important than the other computed ones, as AUC or specificity?

For example:

DHA and EPA both have good accuracy, sensitivity, and specificity values but ~low MCC and AUC values.

The AA metabolite has the highest AUC and MCC values and a lower ACC=0.62 and Specificity=0.45

6/ English language: need to be improved (long sentences).

7/ Titles of the paragraphs could also be improved. A small example: Model testing → Model evaluation? (481)

Reviewer #4 (Remarks to the Author):

This review covers the identification of lipids in this paper (Supplementary data) and in particular whether the data presented backs up the claims made by the authors for the detection of SPMs and related mediators in the samples as shown. Spectra and retention times are compared with the synthetic standards, provided in a handbook on the website of Charles Serhan, Harvard, for reference purposes (<http://serhanlab.bwh.harvard.edu/wp-content/uploads/2019/05/UPDATED-2019-Spectra-Book.pdf>).

Supplementary Figures 9 and 12. These do not appear to be real raw data, it looks like invented chromatograms where most peaks appear to be placed on a flat baseline. It would be essential to request the raw data to ensure the chromatograms match the figures as shown.

Supplementary Figure 1.

Out of the 53 lipids presented, 11 have been presented with chromatograms where the peaks look like respectable peaks (although retention time comparison with standards wasn't provided), 14 are suggestive of lipids being present, but are not sufficiently proven, and 28 are judged to be not present at all by any criteria. This means over 50% of the lipids claimed have not actually been detected at all. Given this, then the whole study (which relies on measurement of these lipids has to be called into doubt). Notably, several of the lipids that are judged to be present are generally abundant mediators that are easily measured by many labs in the world (PGE2, PGD2, LTC4 and others), while the lipids not convincingly measured or absent, are the SPMs and related mediators which many labs worldwide report being unable to measure in biological samples.

When measuring low abundance lipids in a biological sample, there will be a complex mix of oxidized fatty acids coming off a column during a particular time window. Thus, identifying lipids present at low abundance rigorously in such a sample is challenging. For abundant lipids like PGE2, PGD2 etc, proving they are there is relatively easy and the group have identified them fine here, however where this all falls down is the lower abundant lipids. The overall impression is that the quality of MS analysis, and measurement of lower abundant lipids is extremely poor in this study, and that the authors are essentially taking very poor quality analytical data, including background noise and claiming it to be detection of real lipids.

(i) Chromatograms. In relation to retention time, no information has been provided for the standards on the instrument used in the Dalli laboratory, and so comparing retention time with authentic standards is not possible with the information provided. The Serhan handbook uses a different column, however some of the abundant lipids (PGE2, PGD2, PGF2a, LXB4) co-elute exactly as they do in the Serhan lab. However, making this comparison, some other lipids are eluting at retention times that are not the same with a few being up to 0.6 min different: e.g. 20-COOH-LTB4. Given that many lipids elute in some of these individual channels, it is not clear that the right lipid is being integrated in some cases, and how the investigators decided which peak to choose is not shown. Full information on retention time based on standards run under the experimental conditions in this lab should be provided.

The general expectation in the field is that a chromatographic peak should be at least 4-5 times signal:noise and have around data 5-6 points or more for it to be considered a convincing signal on LC/MS/MS. It would be interesting to know what sort of peak the investigators consider to be

LOD when some of these are clearly just baseline noise. The investigators state that they use a criteria of 2000 for the cps value of integrated peaks. However, the background noise level in different MRM channels can vary hugely meaning that this sort of approach could lead to serious errors, and it is not the usual way to determine whether a signal is a real peak or not.

(ii) Spectra: The majority of the spectra do not match those in the Serhan handbook. This is not surprising given how weak the signals are. It would be virtually impossible to get spectra that do match considering the lipids are eluting in biological mixtures where many lipids with the same m/z are either co-eluting or almost co-eluting and also there will be background signals. Thus, the spectra (except for the very abundant lipids, like PGE2 or PGD2) are not very useful in themselves and don't help make the case that these lipids are here.

Below are details on each lipid.

1. RvD1. There are some noisy looking peaks shown that are not convincing. The peak that has been integrated is poor quality, and is borderline. The spectrum looks nothing like the spectrum in the Serhan handbook available online. The large diagnostic ions at m/z 135, 141, 215, are all missing. A higher m/z ion at 339 is present but is in the baseline and not convincing as a diagnostic ion. Ions around 287 and 299 are ignored. It is not possible to conclusively state that this lipid is present in this sample

2. RvD2. A peak is selected in the chromatogram that is beside a larger peak. Quality of the selected peak is relatively poor, with few data points. The spectrum does not support the presence of the lipid, when compared with the spectra handbook as it is missing the ion at m/z 141. There are labels for ions at m/z 215 and 233, but on close inspection one can see that these ions do not exist at all, and likely the same can be said for 313. A major ion is ignored at around 218. The ion at 113 appears to be in the baseline and is not the large ion labeled. The spectrum of the standard shows an ion (unlabeled) around 175, and this is in the sample, however in the absence of the other diagnostic ions, this is not convincing identification. It is not possible to state that this lipid is present in this sample

3. RvD3. A lipid is shown which looks like a small but reasonable peak. However, the spectrum shown does not support the presence of the lipid in the sample. A large diagnostic ion around m/z 219 that this lipid does not have is ignored completely. 101, 181 appear to be missing, despite being labeled. m/z 137, 215, is not the large ion that it is implied to be, but a far smaller one in the noise, 259 seems to be absent, and is instead labeling 260. Several ions at the labels given are present as baseline, not proper signals, e.g. 215. Most signals in the area that has been expanded 5 fold appear like baseline noise blown up, and claiming these are diagnostic is impossible. The chromatogram shows a low peak, but the spectrum is not useful at all.

4. RvD4. It is impossible to say that this is anything other than baseline noise and it cannot be claimed this is a real peak. The spectrum does not match the standard at all, in particular, it is missing ions at 131, 215 and 225, as well as 259. The ion claimed to be 113 is baseline. The standard does not contain an ion at 288, or a large ion at 255 (only a very tiny one). Three large ions at 141, 185 and 188 are totally ignored. The lipid is judged to be absent in this sample.

5. RvD5. The peak is borderline, and in my lab we would not publish this as a real lipid, however it is almost good enough. The spectrum contains several of the ions it should, although having to blow it up by 10 fold means that many of these ions are tiny and potentially nothing more than baseline noise. In terms of S:N this is a borderline peak, and not clearly proven to be present in the sample.

6. RvD6. The retention time of this lipid is later than it should be based on the Serhan book (notably the other lipids match very well the retention time in the book, so this one should also). Perhaps they have integrated the wrong peak. The MS/MS has some ions that seem to be the ones

in the handbook, however given that the spectrum had to be expanded 10-fold assignment of 199 and 279 is unconvincing and looks like baseline noise. The lipid should contain large diagnostic ions at 199 and 279 and there is no way that structure can be assigned based on these background noise peaks presented here. The chromatogram shows a peak, but the spectrum doesn't support the structural assignment as claimed and retention time needs to be shown of standard.

7. 17R-RvD1 (also AT-RvD1). This is not a peak it is a single data point in baseline noise. The spectrum is also extremely poor. Large ions at 135, 141, 233 are totally absent in this spectrum, and the ion at 215 is most likely background noise. The lipid is judged to be absent in this sample.

8. 17R-RvD3 (AT-RvD3). This is not a peak it is a single data point in baseline noise. The spectrum does not match at all the standard, since the claimed ions at 137 and 147 (which should be large ions) are baseline noise here, and large ions at around 175 and 218 are ignored. The ion at 165 is not the one that appears labeled but one that looks like baseline noise. The lipid is judged to be absent in this sample

9. PD1. A weak peak that just about passes detection criteria. However, the spectrum does not match the standard. 137 is missing, 159 is noise, 188 (if it is there, is baseline noise, while in the standard it is a clear ion. 206, 261 (two diagnostic ions in the standard) are missing. The the chromatogram is suggestive, and passes S:N, but the spectrum isn't helping prove the lipid is present.

10. 17R-PD1 (not in the handbook, but expect to have same MS/MS as PD1). The chromatogram is very strange with a peak on the edge of the baseline and then a large broad noisy "peak" just following. I would be concerned about using this for identification because we cannot see the baseline in that channel enough to judge if this is a peak. The spectrum should have large ions at 137 (missing), 153 (baseline), 188, 206, but these are not clearly present or are in the baseline. This lipid might be present but it isn't clearly demonstrated as the size of the baseline noise is not clear.

11. PDX (AT-PD1 in the Serhan handbook). No standard is in the handbook, but this should have same spectrum as PD1 (so I will compare these). The chromatogram is poor and the peak is not a convincing peak at all. Ion at 137 is missing and 153 is very weak. 206 is missing, 217 is weak, and like all the others being expanded 10 fold means that mainly baseline noise is being views as ions. The lipid is judged to be absent in the sample.

12. 22-OH-PD1. There is no peak in this chromatogram, it is just baseline noise. There is no spectrum in the handbook however this spectrum is missing the labeled ions at 199 and 243. The lipid is judged to be absent in the sample.

13. PCTR1: baseline noise is visible and no peak is apparent. The spectrum is nothing like the one in the handbook and there are several ions not labeled at all. The lipid is judged to be absent from the sample

14. PCTR2. A peak that approaches the signal:noise threshold but is very poor quality. It is possible to just about see the minor marks on the axis close up and one can see that the ion labeled 179 is around 175. A large ion at 183 is ignored. Based on the peak shown and S:N the lipid is judged to be absent.

15. PCTR3. The peak passes criteria, however the retention time in the Serhan handbook is 10.2, while this is 9.9, so showing co-elution with a standard would be important. The spectrum doesn't support the structure since the 231 ion is not a large signal but down in baseline noise. When zooming into the spectrum, the markers for the x-axis are a bit shifted, however there is no clear ion at 325 visible. So the spectrum as shown doesn't clearly match the lipid. Borderline peak and

isn't possible to state whether the lipid is conclusively present in the sample based on the data shown.

16. MaR1: The peak is baseline noise, and the spectrum does not match the standard in the handbook, with mission ion at 250. 315 is a baseline ion, not the large one labeled and no ion appears at 323. The lipid is judged to be absent from the sample

17. 7S,14S-diHDHA (an isomer of Mar1), no spectrum in the handbook but it will be the same as Mar1. There is no peak on this chromatogram. This is baseline noise. The spectrum is not confirming the structure as no 123 or 177 present at all, despite being blown up X10. The lipid is judged to be absent from the sample.

18. MaR2; Chromatogram peak is weak and not convincingly above baseline noise. No structure or spectrum available to compare with however there is an absence of any diagnostic ions above baseline noise. The lipid is judged to be absent from the sample

19. 4S,14S-diHDHA: Chromatogram peak is weak and not convincingly above baseline noise. Despite being blown up a massive 20 times, the spectrum does not convincingly show the ions at 159, 187 or 239 as diagnostic ions. The lipid is judged to be absent from the sample

20. 22-OH-MaR1. The peak is very weak and borderline. No reference spectrum is available and the spectrum provided gives no clear structural information. The lipid is not clearly shown to be present in the sample.

21. 14-oxo-MaR1. The peak is not sufficiently above baseline. No reference spectrum is available and the spectrum provided gives no clear structural information. The lipid is judged to be absent from the sample

22. MCTR1. This is baseline noise. The spectrum does not at all look like the reference spectrum (I would not expect it to given there is no peak). The lipid is judged to be absent from the sample

23. MCTR2. A very weak borderline peak that is not convincingly above baseline noise. No ions at 179, 191 or 217 are visible in the spectra. The lipid is judged to be absent from the sample

24. MCTR3. A convincing peak (not a strong one) is shown, but the integration is missing half the peak, so quantification is questionable. The ion at 325 (in the standard) is missing, 191 looks like a baseline ion, and the spectrum does not look like the one in the standard book. The lipid is not clearly shown to be present in the sample.

25. RvT1. An unconvincing peak that is close to baseline. The spectrum is missing the large ion at 211, and is not convincing. The lipid is not clearly shown to be present in the sample.

26. RvT2. Likely baseline noise in the chromatogram is labeled as a peak. The spectrum is very different to the standard spectrum, mission ions at 181, 207, 209, 255, and is not convincing. The lipid is considered absent due to S:N being too low.

27. RvT3. Baseline noise being labeled as a peak and no spectrum is provided in the handbook. Based on the chromatogram, the lipid is judged to be absent from the sample.

28 RvT4. A single data point in noise is being labeled as a peak. The spectrum is not matching the lipid missing visible ions at 143, 193, 199, and with 211 in baseline. The lipid is judged to be absent from the sample.

29. RvD1n-3DPA. Very poor quality peak, that would not pass criteria. The spectrum is not good enough to confirm the lipid, with baseline ions labeled as diagnostic ions and no ion at 233. The

lipid is judged to be absent from the sample.

30. RvD2n-3DPA. Very poor quality peak, that would not pass criteria. No spectrum is available in the handbook. The lipid is judged to be absent from the sample from this chromatogram.

31. RvD5n-3DPA. Very poor quality peak, that would not pass criteria. The spectrum is not good enough to confirm the lipid, with baseline ions labeled as diagnostic ions and no ion at 199 or 201, despite labels being included. 143 is also missing. The lipid is judged to be absent from the sample.

32. PD1n-3DPA. Baseline noise in a chromatogram being labeled as a peak. Spectrum does not match the spectrum in the handbook, missing significant ions at 155, 183, 263. The lipid is judged to be absent from the sample.

33. MaR1n-3DPA. Baseline noise in a chromatogram being labeled as a peak. Spectrum does not match the spectrum in the handbook, missing diagnostic ions (they look like background noise at 143, 223, 233). The lipid is judged to be absent from the sample.

34. RvE1. A weak present is present, around same RT as in standard handbook, however, spectrum is very different to handbook. To be sure this lipid was the correct one, the standard should be run under the same conditions on the same column (this information is not provided). Given how small the peak is it is, it is hard to be convinced this is really properly identified. Borderline but possibly present.

35. RvE2. A weak present is present, around same RT as in standard handbook, however, spectrum is very different to handbook, missing ions at 115, 213. To be sure this lipid was the correct one, the standard should be run under the same conditions on the same column (this information is not provided). Given how small the peak is it is, it is hard to be convinced this is really properly identified. Borderline but possibly present.

36. RvE3. Baseline noise on the side of a noisy peak, which may not be a peak itself is being integrated. Spectrum is not convincingly like the standard. The lipid is judged to be absent from the sample.

37. LXA4. No peak in this chromatogram, just baseline noise. A weak spectrum that is not helping with identification. The lipid is judged to be absent from the sample.

38. LXB4. A peak that looks ok, but is not particularly strong. Spectrum has some diagnostic ions. Presuming that this lipid co-elutes with the standard, then it should be the correct one.

39. 5S,15S-diHETE. A weak shoulder alongside a bigger peak that doesn't pass quality, and is surrounded by several other similar weak peaks. The spectrum has blown up a lot of noise and so deciding that these are diagnostic signals is impossible. Some larger ions that would not be expected (195, 165) are ignored for example. The lipid judged absent.

40. 15-epi-LXA4. Peak is background noise and does not show the lipid to be present. A weak and unconvincing spectrum. The lipid is judged to be absent from the sample.

41. 15-epi-LXB4. Peak is background noise and does not show the lipid to be present. A weak and unconvincing spectrum. The lipid is judged to be absent from the sample.

42. LTB4: The chromatogram and spectrum are convincingly showing the lipid

43. 5S,12S-diHETE: this is noise in the baseline, and is shown as a tiny shoulder attached to a bigger peak. It would be impossible to get a spectrum of this shoulder without the larger adjacent peak being also fragmented. The lipid is judged to be absent from the sample

44. 6-trans-12-epi-LTB4. This is a peak presumably measured in the same channel as LTB4 (unclear), and it would be essential to see a standard run alongside so that RT can be confirmed (this was not shown). No spectrum is available, but comparing with LTB4 there are some similarities. Insufficient information provided but convincing peak shown.
45. 6-trans-LTB4. Potentially there, although needs to be shown alongside standard on chromatogram for retention time. Spectrum is not showing the lipid convincingly. Insufficient information provided but possibly present as peak is convincing.
46. 20-COOH-LTB4. A decent peak, and spectrum looks similar, so possible match although the retention time is very different from the handbook at 0.6min later, so the RT of the standard needs to be provided. Insufficient information provided but potential match as strong peak shown.
47. LTC4. Retention time is a bit early so wonder if the right peak has been integrated. Should be around 10.3, and there is another peak there. But the peak is ok as shown. Spectrum appears reasonable, so probably ok.
48. LTD4. Chromatogram and spectrum are fine
49. LTE4. Chromatogram and spectrum are fine
50. PGD2. Chromatogram and spectrum are fine
51. PGE2. Chromatogram and spectrum are fine
52. PGF2a. Peak is low and borderline, but I suspect this one is ok
53. TXB2. This normally elutes as a sort of blob or double peak, and I suspect the whole area should have been integrated. Instead they have taken a small shoulder at the start and ignored the rest. The lipid is likely to be there, but the quantitation is poor and missing most of the lipid

Reviewer #5 (Remarks to the Author):

Comments on the lipid analyses by LC-MS

Criteria for the positive identification and quantitation of analytes by LC-MS include among other parameters elution at the correct retention time, a "clean-looking" LC peak not co-eluting with nearby compounds, and an acceptable level of signal-to-noise.

The chromatograms illustrated in Supplement Figure A - X and A - I include many analyses that do not achieve the required uncontaminated appearance of the chromatographic peak and many cases in which the peaks exhibit a very low and unacceptable signal-to-noise ratio.

What is an acceptable signal-to-noise ratio? This number should be defined in the methods such that a signal-to-noise below the cut-off be rejected and the analysis deemed invalid. One might suggest that a low (non-stringent) cut-off be, for example, 5:1 signal:noise. Many of the representative chromatograms in the Supplement do not achieve even such a non-stringent criterion and very clearly should be rejected in any generally acceptable and validated LC-MS methodology.

In each case in the relevant Supplement chromatograms the investigators include a mass spectrum (or product ion spectrum) that gives some support to the designated peak identity.

However, this cannot overrule a contaminated chromatographic peak and/or an unacceptably low signal-to-noise ratio. When the signal-to-noise ratio is 1:1 or near that, one would question what is the appearance of the mass spectra in nearby areas of the chromatogram, (the illustrated chromatograms are already selected for the designated molecular ion)? Also, to what extent do the mass spectra fragmentations exactly match the standard – what leeway is taken in this parameter?

[Minor, typo: Main manuscript page 17, 4th line from bottom: It appears unlikely the retention time match could be restricted to 0.05 sec as stated].

Point by point response to reviewer comments:

Reviewer #1 (Remarks to the Author):

The authors have dropped the murine data from the manuscript and this corrects one of the major problems with the manuscript.

For one used to looking at bar graphs or curves when discussing changes in concentration of various reactants in patients before and after treatment it is a little difficult to follow the graphs as noted in Figs. 2 and 4. Perhaps a principal component analysis could be done comparing the four different groups (non-responders and responders at baseline, non-responders and responders after treatment). The point the authors make is valid and interesting (really the point of the entire work) but the changes are difficult to make out with the graphics and analyses used.

We thank the reviewer for their comment. Given that the aim of our analysis was to assess whether there are differences in lipid mediator levels between the two patient groups at each of the intervals we do not think that performing multivariate analysis with data from all the four groups will add much to the current manuscript. We instead have included results comparing the concentrations of the different lipid mediator families between the two groups. These new results are presented in figures 1, 3 and 4 discussed on page 5 line 109-112, page 11 pages 9-10 lines 214-236 and page 11 lines 247-260. Furthermore, the individual mediator values are provided in Supplemental Tables 3, 5 and 7.

Finally, this reviewer has a minor quibble with the sentence added in page 3 lines 54-56. The point here was that CD39 levels on Tregs is a biomarker that predicts MTX responses and CD39 is critical for the formation of adenosine in the extracellular space (one of the mechanisms described in the cited review).

We have revised this line to clarify the identified mechanism of action as detailed below and on page 3 lines 58-61:

*Furthermore, a recent study found that the expression CD39 in peripheral blood regulatory T-cells is important for the observed beneficial actions of MTX. Whereby, patients that displayed a lower density of this receptor on peripheral blood regulatory T-cells were unresponsive to MTX*⁸,

Reviewer #2 (Remarks to the Author):

The authors have been very responsive to the reviewer's comments and provided new data and analysis. Concerns about experiments that were aimed at defining a mechanism for MTX regulation of 15-LOX or SPM formation have been addressed by not including the data. Even though this eliminates a major weakness of the central hypothesis it also significantly lowers the potential impact of this study. The study is now limited to an elegant but descriptive lipidomic analysis but provides no logical mechanism to explain the coincident changes in plasma levels of selected SPM, a conclusion that is based on machine learning analysis. The changes in plasma SPM may have multiple and indirect causes, which is a significant limitation of the study.

We thank the reviewer for their comments. As detailed in our prior response the removal of the mouse data was indicated by the editors as a prerequisite to our resubmission.

Reviewer #3 (Remarks to the Author):

The actual version of the manuscript is improved but needs more clarifications and ameliorations.

I have few comments:

1/ If possible to specify that you used the linear SVM.

We thank the reviewer for the suggestions. We have revised the manuscript to clarify the text using the indicated methodologies as detailed on page 19 lines 463 of the revised manuscript and below:

“A nonlinear kernels radial basis function SVM was used.”

2/ Dimension of the input dataset: better to include Number of individuals x Number of variables

We thank the reviewer for their suggestion and have included this information in the revised Table 1.

3/ RandomForest: when hyper tuning the parameters, you fixed mtree=10000 because it gave you the best performing models when running all the experiments. What about the number of variables, did you also use the same number for training all the models?

We thank the reviewer for raising this question. We have revised the manuscript to clarify the optimization step of mtry as detailed on page 20 lines 473 - 479 of the revised manuscript and below:

‘In the present studies using the R package “randomForest” (<https://cran.r-project.org/package=randomForest>), which uses bootstrapping as the test method, we first used a small loop to test the different mtry values, the number of variables randomly sampled as candidates at each split. Then using the mtry value that gave the best classification performance for each model we tested a number of ntree values, the number of decision trees that are created before creating the consensus classifier tree, that gave the most stable models.

4/ Table 1: Could you please specify the formula of MCC you used? How exactly did you compute it using the mltools? Was it based on the values of TP, FP, TN and FN? If yes, it is possible to compute them manually and you will obtain different results.

Thank you for this question, the formula used is detailed below.

$$MCC = \frac{TP \times TN - FP \times FN}{\sqrt{(TP + FP)(TP + FN)(TN + FP)(TN + FN)}}$$

Given that the results from this analysis only additive to that obtained with the ROC analysis suggested by the reviewer, in order to simplify the manuscript we have opted to leave out this data.

The MCC values obtained from running the models on Cohort 1 are different between the first and the revised version, but we have the same values for the other evaluation metrics, why? As far as I understood, Cohort 1 didn't change, right?

How is it possible to have high mean sensitivity and specificity values and quite low AUC value,

as for the DHA model?

Thank you raising this question. The apparent discordance arises from the way we chose to present the data where the results for Cohort 1 and Cohort 2 were displayed side by side without highlighting the origin. We have revised Table 1 to clarify the origin of the data. The AUC values were obtained using lipid mediator concentrations from Cohort 2 (test dataset) and not from Cohort 1 (random fraction separated during the training step for the evaluation process of the model) as can now be appreciated in the revised table.

5/ Most relevant biomarkers: You selected the best metabolites DHA and EPA based mainly on the values of accuracy? Why this metric is more important than the other computed ones, as AUC or specificity?

For example:

DHA and EPA both have good accuracy, sensitivity, and specificity values but ~low MCC and AUC values.

The AA metabolite has the highest AUC and MCC values and a lower ACC=0.62 and Specificity=0.45

We thank the reviewer for raising this important point. This point is discussed on pages 7 lines 132-142 and page 8 lines 148-158. We have clarified the selection criteria in the revised manuscript. Briefly, in our initial assessment we selected as good models those that showed a consensus in the % accuracy score and AUC in both random forest and SVM methodologies for the distinct lipid mediator families. In order to better understand which of the mediators from the distinct mediator families were indeed important in predicting outcome the most relevant biomarkers were selected based in the "importance" function of randomForest, which shows the relevant variables in the performance of the machine learning models.

6/ English language: need to be improved (long sentences).

We thank the reviewer, we have revised the document to improve readability.

7/ Titles of the paragraphs could also be improved. A small example: Model testing —> Model evaluation? (481)

We have revised the title as suggested.

Reviewer #4 (Remarks to the Author):

This review covers the identification of lipids in this paper (Supplementary data) and in particular whether the data presented backs up the claims made by the authors for the detection of SPMs and related mediators in the samples as shown. Spectra and retention times are compared with the synthetic standards, provided in a handbook on the website of Charles Serhan, Harvard, for reference purposes (<http://serhanlab.bwh.harvard.edu/wp-content/uploads/2019/05/UPDATED-2019-Spectra-Book.pdf>).

We thank the reviewer for their efforts and detailed evaluation. While the spectra book available on Dr Serhan's website is an invaluable resource for the identification for lipid mediators, in particular SPM, it is well known in the field that MS/MS fragmentation patterns, in particular ion intensities vary from instrument to instrument and from one instrument line to another. Curiously this aspect is noted by the reviewer in their mark on the RvE1 spectrum in point 34 below, however it appears to be ignored for the rest of the mediators.

Therefore, the ion intensities observed in this spectra book are primarily relevant to Sciex 5500 instruments. Whereas, and as detailed in the methods section, our data was obtained using a Sciex 6500+ instrument. In addition, and potentially more important, it is well established that there is an important matrix effect on fragmentation patterns of lipid mediators. Indeed, it is widely appreciated that while key diagnostic fragments are retained, the presence of a matrix leads to changes in fragmentation patterns and ion ratios compared to when the matrix is absent. This phenomenon is well documented in the analytical literature and reviewed in the following publication (PMID: 18680189). Thus, while the fragments reported in Dr Serhan's spectra book are relevant to mediator identification, the ion intensities observed herein are not, given that these spectra were obtained using a different instrument line and in the absence of matrix. In order to substantiate this argument and to facilitate the evaluation of the MS/MS spectra obtained for identification of mediators in our samples we have provided reference MS/MS fragmentation spectra that we obtained either in phase (methanol/water) or by spiking an equal amount of the reference mediators in matrix. This is now presented in Appendix A. Furthermore, we have included reference spectra obtained in matrix in the revised manuscript to facilitate the evaluation of the MS/MS spectra used for the identification of lipid mediators from patient samples. These can be found in Supplemental Figures 1-4 of the revised manuscript.

Supplementary Figures 9 and 12. These do not appear to be real raw data, it looks like invented chromatograms where most peaks appear to be placed on a flat baseline. It would be essential to request the raw data to ensure the chromatograms match the figures as shown.

Indeed, these are not raw data but a representation of chromatograms as is widely used in the literature. We now provide representative chromatograms and the reference standards used to identify each of the enantiomers of interest. These can be found in Supplemental figure 9-12.

Supplementary Figure 1.

Out of the 53 lipids presented, 11 have been presented with chromatograms where the peaks look like respectable peaks (although retention time comparison with standards wasn't

provided), 14 are suggestive of lipids being present, but are not sufficiently proven, and 28 are judged to be not present at all by any criteria. This means over 50% of the lipids claimed have not actually been detected at all. Given this, then the whole study (which relies on measurement of these lipids has to be called into doubt). Notably, several of the lipids that are judged to be present are generally abundant mediators that are easily measured by many labs in the world (PGE2, PGD2, LTC4 and others), while the lipids not convincingly measured or absent, are the SPMs and related mediators which many labs worldwide report being unable to measure in biological samples.

We respectfully disagree with the statement that more than 50% of the mediators are not present at all. As detailed in the point by point response below, all the mediators reported in the present studies were identified in accordance with widely published and accepted criteria. Furthermore, we are unaware of the claim that many labs worldwide cannot measure these molecules. We would like to highlight just some publications where SPM have been identified and quantified in biological systems using criteria we have illustrated, this includes: PMID: 25139986; PMID: 29206462; PMID: 31492430, PMID: 29789130; PMID: 27461565; PMID: 27656142; PMID: 26324767; PMID: 25049337, PMID: 26180051.

When measuring low abundance lipids in a biological sample, there will be a complex mix of oxidized fatty acids coming off a column during a particular time window. Thus, identifying lipids present at low abundance rigorously in such a sample is challenging. For abundant lipids like PGE2, PGD2 etc, proving they are there is relatively easy and the group have identified them fine here, however where this all falls down is the lower abundant lipids. The overall impression is that the quality of MS analysis, and measurement of lower abundant lipids is extremely poor in this study, and that the authors are essentially taking very poor quality analytical data, including background noise and claiming it to be detection of real lipids.

We respectfully disagree with this comment. As illustrated in the response above and in the supplemental figures as well as in the responses below, the chromatograms used for establishing retention time fulfill the minimum criteria, furthermore in the MS/MS spectra we identify at least 6 diagnostic ions with 1 or more being backbone fragments.

(i) Chromatograms. In relation to retention time, no information has been provided for the standards on the instrument used in the Dalli laboratory, and so comparing retention time with authentic standards is not possible with the information provided. The Serhan handbook uses a different column, however some of the abundant lipids (PGE2, PGD2, PGF2a, LXB4) co-elute exactly as they do in the Serhan lab. However, making this comparison, some other lipids are eluting at retention times that are not the same with a few being up to 0.6 min different: e.g. 20-COOH-LTB4. Given that many lipids elute in some of these individual channels, it is not clear that the right lipid is being integrated in some cases, and how the investigators decided which peak to choose is not shown. Full information on retention time based on standards run under the experimental conditions in this lab should be provided.

We have provided full information of retention times in the revised manuscript for each of the mediators identified. In addition, we also provided detailed information in the methods section on the criteria used for peak selection as detailed on page 18 line 416-423 of the manuscript and detailed below:

'Each lipid mediator was identified using established criteria, these included: 1) presence of a peak with a minimum area of 2000 counts, 2) matching retention time to synthetic or authentic standards with maximum drift between the expected retention time and the observed retention time of 0.05 seconds 3) >4 data points and 4) matching of at least 6 diagnostic ions to that of reference standard, with a minimum of one backbone fragment being identified^{13, 20, 36}.

Calibration curves were obtained for each mediator using lipid mediator mixtures at 0.78, 1.56, 3.12, 6.25, 12.5, 25, 50, 100, and 200 pg that gave linear calibration curves with an r^2 values of 0.98– 0.99.'

The general expectation in the field is that a chromatographic peak should be at least 4-5 times signal:noise and have around data 5-6 points or more it for it to be considered a convincing signal on LC/MS/MS. It would be interesting to know what sort of peak the investigators consider to be LOD when some of these are clearly just baseline noise. The investigators state that they use a criteria of 2000 for the cps value of integrated peaks. However, the background noise level in different MRM channels can vary hugely meaning that this sort of approach could lead to serious errors, and it is not the usual way to determine whether a signal is a real peak or not.

This information is now provided for each of the mediators identified in the point by point response below. As one can observe these criteria are fulfilled with peaks having at least 5 data points and a signal to noise ratio of 4.

(ii) Spectra: The majority of the spectra do not match those in the Serhan handbook. This is not surprising given how weak the signals are. It would be virtually impossible to get spectra that do match considering the lipids are eluting in biological mixtures where many lipids with the same m/z are either co-eluting or almost co-eluting and also there will be background signals. Thus, the spectra (except for the very abundant lipids, like PGE2 or PGD2) are not very useful in themselves and don't help make the case that these lipids are here.

As stated in the response, the main reasons for these differences observed between the spectra presented in our manuscript and those from the Serhan spectra book are: 1) that these were obtained using different instrument lines and 2) in the absence of a matrix. The matrix contribution is a critical factor and we illustrate this in Appendix A where we provide a side-by-side comparison of MS/MS spectra obtained for the same amount of mediator in matrix and in phase. Therefore we are confident that the identification criteria employed are robust and reflect the presence of the mediator of interest in the samples.

Below are details on each lipid.

1. RvD1. There are some noisy looking peaks shown that are not convincing. The peak that has

been integrated is poor quality, and is borderline. The spectrum looks nothing like the spectrum in the Serhan handbook available online. The large diagnostic ions at m/z 135, 141, 215, are all missing. A higher m/z ion at 339 is present but is in the baseline and not convincing as a diagnostic ion. Ions around 287 and 299 are ignored. It is not possible to conclusively state that this lipid is present in this sample.

Lipid mediators have several natural isomers that result in the chromatogram looking 'noisy'. This is true for the classic eicosanoids were for example LTB_4 has 2 natural isomers: 5S, 12S-diHETE, and Δ^6 trans, 12-epi LTB_4 , which elute very close to the mediator. As noted in Supplemental figure 1A and Supplemental Table 2 the peak fulfills all the minimum criteria identified by the reviewer in their above statement for positive identification i.e., matching retention time, signal to noise ratio of 12 and 7 data points. In regards to the MS/MS spectrum, we now provide a reference MS/MS spectrum obtained in Matrix which was used as reference, while indeed some of the fragments are missing, more than 6 diagnostic ions (8 ions) were identified in the sample that included 3 backbone fragments. Therefore, we disagree with the comment that it is not possible to state whether the mediator is present in the sample.

2. RvD2. A peak is selected in the chromatogram that is beside a larger peak. Quality of the selected peak is relatively poor, with few data points. The spectrum does not support the presence of the lipid, when compared with the spectra handbook as it is missing the ion at m/z 141. There are labels for ions at m/z 215 and 233, but on close inspection one can see that these ions do not exist at all, and likely the same can be said for 313. A major ion is ignored at around 218. The ion at 113 appears to be in the baseline and is not the large ion labeled. The spectrum of the standard shows an ion (unlabeled) around 175, and this is in the sample, however in the absence of the other diagnostic ions, this is not convincing identification. It is not possible to state that this lipid is present in this sample

The peak in the chromatogram initially provided fulfilled the minimum criteria illustrated above, with matching retention time, signal to noise ratio of 4 and 5 data points. Furthermore, the adjacent peak was likely an isomer as illustrated above. For clarity we provided an alternative chromatogram obtained from the samples being analyzed, the peak in this chromatogram in addition to matching the retention time, has signal to noise ratio of 8 and 9 data points. As to the MS/MS spectrum, we now provide a reference spectrum obtained in matrix. While indeed some of the ions are indeed missing, likely due to an interference of the matrix in the fragmentation process, a phenomenon that is well appreciated to influence fragmentation in collision-induced dissociation, there are more than the minimum 6 diagnostic ions required for identification (11 ions) including 3 backbone fragments. Therefore, we disagree with the comment that it is not possible to state whether the mediator is present in the sample. This information is provided in Supplemental Figure 1B and Supplemental Table 2.

3. RvD3. A lipid is shown which looks like a small but reasonable peak. However, the spectrum shown does not support the presence of the lipid in the sample. A large diagnostic ion around m/z 219 that this lipid does not have is ignored completely. 101, 181 appear to be missing,

despite being labeled. m/z 137, 215, is not the large ion that it is implied to be, but a far smaller one in the noise, 259 seems to be absent, and is instead labeling 260. Several ions at the labels given are present as baseline, not proper signals, e.g. 215. Most signals in the area that has been expanded 5 fold appear like baseline noise blown up, and claiming these are diagnostic is impossible. The chromatogram shows a low peak, but the spectrum is not useful at all.

We would like to point out that the chromatogram fulfills all the required criteria for positive identification in that the retention time matches that of the standard, the signal to noise ratio is of 6 and the number of data points are 9. As to the reason why the large 219 ion is not included as part of our identification in the reference spectrum is because we have, to date, not been able to assign this ion to any specific fragment and therefore we do not feel comfortable including it as part of our diagnostic criteria. As previously illustrated, the absence of some of the ions from the samples is likely due to interference of the matrix in the fragmentation process coupled with the relatively low abundance of the mediator in the sample. Despite this, the MS/MS spectrum carries 8 diagnostic ions with 1 being backbone fragments. This information is provided in Supplemental Figure 1C and Supplemental Table 2.

4. RvD4. It is impossible to say that this is anything other than baseline noise and it cannot be claimed this is a real peak. The spectrum does not match the standard at all, in particular, it is missing ions at 131, 215 and 225, as well as 259. The ion claimed to be 113 is baseline. The standard does not contain an ion at 288, or a large ion at 255 (only a very tiny one). Three large ions at 141, 185 and 188 are totally ignored. The lipid is judged to be absent in this sample.

We acknowledge that the quality of the chromatogram for this mediator was suboptimal. This was likely due to the file conversion, which reduced the quality of the image. We have addressed this issue and replaced the image with a better quality. In this chromatogram we would like to highlight the matching retention time, a signal to noise ratio of 9 and the fact that the peak is composed of 11 data points. We have also provided the appropriate reference spectrum for this mediator and as one can appreciate there are 10 diagnostic ions including 2 direct backbone fragments to confirm presence of RvD4. This information is provided in Supplemental Figure 1D and Supplemental Table 2.

5. RvD5. The peak is borderline, and in my lab we would not publish this as a real lipid, however it is almost good enough. The spectrum contains several of the ions it should, although having to blow it up by 10 fold means that many of these ions are tiny and potentially nothing more than baseline noise. In terms of S:N this is a borderline peak, and not clearly proven to be present in the sample.

We thank the reviewer for their providing insights into what they would consider a real peak. We would like to note that the peak fulfills the minimum criteria that they stated would be required for positive identification. As to the comment regarding the ions being tiny and therefore being noise we would like to point out that the ion intensity of fragments is relative to the parent ion and no real measure of abundance as also highlighted by the fact that the same region in the reference spectrum is amplified x5. This information is provided in Supplemental Figure 1E and

Supplemental Table 2.

6. RvD6. The retention time of this lipid is later than it should be based on the Serhan book (notably the other lipids match very well the retention time in the book, so this one should also). Perhaps they have integrated the wrong peak. The MS/MS has some ions that seem to be the ones in the handbook, however given that the spectrum had to be expanded 10-fold assignment of 199 and 279 is unconvincing and looks like baseline noise. The lipid should contain large diagnostic ions at 199 and 279 and there is no way that structure can be assigned based on these background noise peaks presented here. The chromatogram shows a peak, but the spectrum doesn't support the structural assignment as claimed and retention time needs to be shown of standard.

The reviewer is correct, we mistakenly included the wrong information. This error is rectified in the revised figures.

While the MS/MS spectrum employed in the identification of RvD6 does not carry all the ions found in the reference spectrum, we identified 10 fragments that correspond to those in the reference spectrum, 2 of which are direct backbone fragments. Once again, the relative abundance of the ions does not rule out their diagnostic utility. It should be also noted that as part of the identification methodologies we only used fragments that appear after centroiding the MS/MS spectral data. This process helps to eliminate any ions that appear as a result of electronic noise. This information is provided in Supplemental Figure 1F and Supplemental Table 2.

7. 17R-RvD1 (also AT-RvD1). This is not a peak it is a single data point in baseline noise. The spectrum is also extremely poor. Large ions at 135, 141, 233 are totally absent in this spectrum, and the ion at 215 is most likely background noise. The lipid is judged to be absent in this sample.

As can be appreciated from the information provided in the chromatogram and Supplemental Table 1 the peak fulfills the criteria stated by the reviewer, in that the retention time is a match to the standard, the signal to noise ratio is of 6 and there are 6 data points in the peak. Therefore, we respectfully disagree with their conclusion. The MS/MS spectrum provided has 9 diagnostic ions one of which one is a direct backbone fragment. Therefore, we refer the reviewer to the response given above regarding the diagnostic utility of the MS/MS spectrum used. This information is provided in Supplemental Figure 1G and Supplemental Table 2.

8. 17R-RvD3 (AT-RvD3). This is not a peak it is a single data point in baseline noise. The spectrum does not match at all the standard, since the claimed ions at 137 and 147 (which should be large ions) are baseline noise here, and large ions at around 175 and 218 are ignored. The ion at 165 is not the one that appears labeled but one that looks like baseline noise. The lipid is judged to be absent in this sample

As can be appreciated from the information provided in the chromatogram and Supplemental Table 1 the peak fulfills the criteria stated by the reviewer in that the retention time is a match to

the standard, the signal to noise ratio is of 5 and there are 8 data points in the peak. Therefore, we respectfully disagree with their conclusion. The MS/MS spectrum provided has 10 diagnostic ions one of which one is a direct backbone fragment. Thus, we also refer the reviewer to the response given above regarding the diagnostic utility of the MS/MS spectrum used. This information is provided in Supplemental Figure 1H and Supplemental Table 2.

9. PD1. A weak peak that just about passes detection criteria. However, the spectrum does not match the standard. 137 is missing, 159 is noise, 188 (if it is there, is baseline noise, while in the standard it is a clear ion. 206, 261 (two diagnostic ions in the standard) are missing. The the chromatogram is suggestive, and passes S:N, but the spectrum isn't helping prove the lipid is present.

As the reviewer confirms the peak passes the identification criteria used in the field. Therefore, this by itself should suffice as a positive identification given that this is was is generally used in the eicosanoid field for identification. Furthermore, as stated in the responses above the MS/MS spectrum used by the reviewer is not the correct reference, we have provided the MS/MS spectrum we employed as a reference which as can be appreciated fulfills the identification criteria described above given the presence of 9 diagnostic ions of which one is a direct backbone fragment. This information is provided in Supplemental Figure 1I and Supplemental Table 3.

10. 17R-PD1 (not in the handbook, but expect to have same MS/MS as PD1). The chromatogram is very strange with a peak on the edge of the baseline and then a large broad noisy "peak" just following. I would be concerned about using this for identification because we cannot see the baseline in that channel enough to judge if this is a peak. The spectrum should have large ions at 137 (missing), 153 (baseline), 188, 206, but these are not clearly present or are in the baseline. This lipid might be present but it isn't clearly demonstrated as the size of the baseline noise is not clear.

We find ourselves in disagreement with the reviewer. The peak fulfills all the identification criteria as illustrated in Supplemental Figure 1J and Supplemental Table 2 in that the retention time is a match to the standard, the signal to noise ratio is of 5 and there are 5 data points in the peak. As regards to the comments on the ion intensities of specific ions we refer the reviewer to the points made above on the influence of matrix on fragmentation patters. We have also provided the reference MS/MS spectrum employed in the identification of this mediator. As can be noted the MS/MS spectrum provided fulfills the minimum criteria in that there are 10 diagnostic ions of which 2 are backbone fragments.

11. PDX (AT-PD1 in the Serhan handbook). No standard is in the handbook, but this should have same spectrum as PD1 (so I will compare these). The chromatogram is poor and the peak is not a convincing peak at all. Ion at 137 is missing and 153 is very weak. 206 is missing, 217 is weak, and like all the others being expanded 10 fold means that mainly baseline noise is being views as ions. The lipid is judged to be absent in the sample.

We wish to point out that PDX is not AT-PD1, indeed these mediators elute 0.7 minutes apart. As can be appreciated in Supplemental Figure 1K and Supplemental Table 2 the peak fulfills the criteria for positive identification in that the retention time is a match to the standard, the signal to noise ratio is of 4 and there are 5 data points in the peak. In addition, we refer the reviewer to the points above discussing the diagnostic utility of the MS/MS spectrum provided. The fact that a portion of the spectrum is zoomed is conducted to facilitate evaluation and does not detract from the diagnostic utility of the fragments. The MS/MS spectrum for this mediator fulfills the minimum criteria for identification give that it presents 9 diagnostic ions, of which one is a direct backbone fragment.

12. 22-OH-PD1. There is no peak in this chromatogram, it is just baseline noise. There is no spectrum in the handbook however this spectrum is missing the labeled ions at 199 and 243. The lipid is judged to be absent in the sample.

We respectfully disagree with the reviewer the peak meets the identification criteria given that the S/N ratio is of 31 and there are 11 data points in the chromatogram. Similarly, there are 10 ions in the MS/MS spectrum of which 3 are backbone fragments. This information is provided in Supplemental Figure 1L and Supplemental Table 2.

13. PCTR1: baseline noise is visible and no peak is apparent. The spectrum is nothing like the one in the handbook and there are several ions not labeled at all. The lipid is judged to be absent from the sample.

We respectfully disagree with the reviewer the peak meets the identification criteria in that the S/N ratio is of 7 and the number of data points is of 6. Furthermore, there are 6 diagnostic ions in the MS/MS spectrum with 2 of the ions being backbone fragment. This information is provided in Supplemental Figure 1M and Supplemental Table 2.

14. PCTR2. A peak that approaches the signal:noise threshold but is very poor quality. It is possible to just about see the minor marks on the axis close up and one can see that the ion labeled 179 is around 175. A large ion at 183 is ignored. Based on the peak shown and S:N the lipid is judged to be absent.

We respectfully disagree with the reviewer. The peak meets the identification criteria, in that the retention time is a match, the S/N ration is of 6 and there are 6 data points in the chromatogram. The MS/MS spectrum also meets identification criteria in that there are 8 diagnostic ions 3 of which are direct backbone fragments. As to the 183 peak, this is not denoted given that we do not have not been able to assign it to a specific fragment of the molecule and therefore do not feel comfortable to include this as part of our identification. This information is provided in Supplemental Figure 1N and Supplemental Table 2.

15. PCTR3. The peak passes criteria, however the retention time in the Serhan handbook is 10.2, while this is 9.9, so showing co-elution with a standard would be important. The spectrum

doesn't support the structure since the 231 ion is not a large signal but down in baseline noise. When zooming into the spectrum, the markers for the x-axis are a bit shifted, however there is no clear ion at 325 visible. So the spectrum as shown doesn't clearly match the lipid. Borderline peak and isn't possible to state whether the lipid is conclusively present in the sample based on the data shown.

As per request we provided the reference chromatogram which demonstrates co-elution. For the reasons detailed in the previous responses we disagree that the 231 ion is not diagnostically useful and have provided a reference MS/MS spectrum in support of our argument. This information is provided in Supplemental Figure 1O and Supplemental Table 2.

16. MaR1: The peak is baseline noise, and the spectrum does not match the standard in the handbook, with mission ion at 250. 315 is a baseline ion, not the large one labeled and no ion appears at 323. The lipid is judged to be absent from the sample

We respectfully disagree with the reviewer. The peak fulfills the identification criteria in that the retention time is a match, the S/N ratio is of 6 and the peak is composed of 5 data points as can be observed in Supplemental Figure 1P and Supplemental Table 2. Furthermore, the MS/MS spectrum provides similar fragmentation patterns to the reference spectrum in matrix with 9 diagnostic ions identified of which 3 being direct backbone fragments.

17. 7S,14S-diHDHA (an isomer of Mar1), no spectrum in the handbook but it will be the same as Mar1. There is no peak on this chromatogram. This is baseline noise. The spectrum is not confirming the structure as no 123 or 177 present at all, despite being blown up X10. The lipid is judged to be absent from the sample.

We respectfully disagree with the reviewer, the peak fulfills identification criteria (matching retention time, peak is composed of 11 data points and the S/N ratio is of 5) as can be observed in Supplemental Figure 1T and Supplemental Table 2, furthermore the absence of two fragments, which are neutral loss ions does not rule out presence of the mediator in light of the fact that there are 9 diagnostic ions in the spectrum 3 of which are direct backbone fragments.

18. MaR2; Chromatogram peak is weak and not convincingly above baseline noise. No structure or spectrum available to compare with however there is an absence of any diagnostic ions above baseline noise. The lipid is judged to be absent from the sample

We respectfully disagree with the reviewer, the peak (matching retention time, the peak is composed of 7 data points and signal to noise ratio is of 30) and MS/MS fragmentation spectrum (9 diagnostic ions, 3 backbone fragments) fulfill identification criteria as can be observed in Supplemental Figure 1Q and Supplemental Table 2.

19. 4S,14S-diHDHA: Chromatogram peak is weak and not convincingly above baseline noise. Despite being blown up a massive 20 times, the spectrum does not convincingly show the ions at 159, 187 or 239 as diagnostic ions. The lipid is judged to be absent from the sample

We respectfully disagree with the reviewer, the peak (matching retention time, the peak is composed of 8 data points and S/N ratio is of 11) and MS/MS fragmentation spectrum (8 diagnostic ions, 1 backbone fragment) fulfill identification criteria as can be observed in Supplemental Figure 1U and Supplemental Table 2.

20. 22-OH-MaR1. The peak is very weak and borderline. No reference spectrum is available and the spectrum provided gives no clear structural information. The lipid is not clearly shown to be present in the sample.

We respectfully disagree with the reviewer, the peak (matching retention time, the peak is composed of 9 data points and S/N ratio is of 6) and MS/MS fragmentation spectrum (9 diagnostic ions, of which 2 are direct fragments) fulfill identification criteria as can be observed in Supplemental Figure 1R and Supplemental Table 2.

21. 14-oxo-MaR1. The peak is not sufficiently above baseline. No reference spectrum is available and the spectrum provided gives no clear structural information. The lipid is judged to be absent from the sample

We respectfully disagree with the reviewer, the peak (matching retention time, the peak is composed of 5 data points and signal to noise ratio is of 23) and MS/MS fragmentation spectrum (8 diagnostic ions, 3 of which are backbone fragments) fulfill identification criteria as can be observed in Supplemental Figure 1S and Supplemental Table 2.

22. MCTR1. This is baseline noise. The spectrum does not at all look like the reference spectrum (I would not expect it to given there is no peak). The lipid is judged to be absent from the sample

We respectfully disagree with the reviewer, the peak (matching retention time, the peak is composed of 6 data points and S/N ratio is of 17) and MS/MS fragmentation spectrum (6 diagnostic ions, 4 of which are backbone fragments) fulfill identification criteria as can be observed in Supplemental Figure 1V and Supplemental Table 2.

23. MCTR2. A very weak borderline peak that is not convincingly above baseline noise. No ions at 179, 191 or 217 are visible in the spectra. The lipid is judged to be absent from the sample

The peak fulfills all identification criteria (matching retention time, the peak is composed of 5 data points and S/N ratio is of 5). Furthermore, the MS/MS spectrum contains 7 diagnostic ions including two of the ones listed in the reviewer's comment above and 3 backbone fragments. This information is provided in Supplemental Figure 1W and Supplemental Table 2.

24. MCTR3. A convincing peak (not a strong one) is shown, but the integration is missing half the peak, so quantification is questionable. The ion at 325 (in the standard) is missing, 191

looks like a baseline ion, and the spectrum does not look like the one in the standard book. The lipid is not clearly shown to be present in the sample.

We respectfully disagree about the integration and quantification, given that as can be clearly appreciated the area that is not integrated is likely to be a shoulder and therefore likely to be a different mediator that elutes closely to MCTR3. With regards to the missing 325 ion, there must be some misunderstanding given that this is clearly labeled in the spectrum provided. As to the low abundance of the 191 ion the referee is directed to the reference spectrum provided in the revised Supplemental Figure 1X and Supplemental Table 2.

25. RvT1. An unconvincing peak that is close to baseline. The spectrum is missing the large ion at 211, and is not convincing. The lipid is not clearly shown to be present in the sample.

We respectfully disagree with the reviewer, the peak (matching retention time, the peak is composed of 6 data points and S/N ratio is of 13) and MS/MS fragmentation spectrum (8 diagnostic ions, of which 2 is a backbone fragments) fulfill identification criteria as can be observed in Supplemental Figure 2A and Supplemental Table 2. The referee is also directed to the response above in relation to the relative abundance of 211 ion.

26. RvT2. Likely baseline noise in the chromatogram is labeled as a peak. The spectrum is very different to the standard spectrum, missing ions at 181, 207, 209, 255, and is not convincing. The lipid is considered absent due to S:N being too low.

It is unclear how the referee calculated the S:N ratio, given that we find that this value is of 6, and therefore well above the accepted threshold as detailed in Supplemental Figure 2B and Table 1. Furthermore, there are 8 ions in the MS/MS spectrum including the presence of 2 backbone fragments, thus this fulfills the criteria outlined previously.

27. RvT3. Baseline noise being labeled as a peak and no spectrum is provided in the handbook. Based on the chromatogram, the lipid is judged to be absent from the sample.

We respectfully disagree with the reviewer, the peak (matching retention time, the peak is composed of 9 data points and S/N ratio is of 8) and MS/MS fragmentation spectrum (9 diagnostic ions of which 3 are backbone fragments) fulfill identification criteria as can be observed in Supplemental Figure 2C and Supplemental Table 2.

28 RvT4. A single data point in noise is being labeled as a peak. The spectrum is not matching the lipid missing visible ions at 143, 193, 199, and with 211 in baseline. The lipid is judged to be absent from the sample.

We respectfully disagree with the reviewer, the peak (matching retention time, the peak is composed of 6 data points and S/N ratio is of 7) and MS/MS fragmentation spectrum (>6 diagnostic ions, 1 or more direct fragments) fulfill identification criteria as can be observed in

Supplemental Figure 2D and Supplemental Table 2. Furthermore, the 143 ion listed by the reviewer as absent, is indeed present in the spectrum.

29. RvD1n-3DPA. Very poor quality peak, that would not pass criteria. The spectrum is not good enough to confirm the lipid, with baseline ions labeled as diagnostic ions and no ion at 233. The lipid is judged to be absent from the sample.

It is unclear how the reviewer has determined that this would not pass criteria given that the peak fulfills the stated criteria with a matching retention time, the peak is composed of 5 data points and the S/N ratio is of 4. Furthermore, we have included an additional peak that as can be appreciated from the results presented in Supplemental Figure 2E and Supplemental Table 2 also clearly fulfills these criteria (matching retention time, the peak is composed of 9 data points and the S/N ratio is of 11). As to the 233 ion, being a *non-ion*, we are in disagreement given that as can be appreciated from the reference spectrum ion intensities for this ion is similar between that obtained in the sample and that obtained in the reference spectrum.

30. RvD2n-3DPA. Very poor quality peak, that would not pass criteria. No spectrum is available in the handbook. The lipid is judged to be absent from the sample from this chromatogram.

It is unclear to us how the reviewer determined that this peak would not pass criteria given that as can be observed in Supplemental Figure 2F and Supplemental Table 2 the peak presented meets the criteria outlined by the reviewer in their comment above in that the retention time is a match with that of the standard, the peak is composed of 11 data points and the S/N ratio is of 7. Furthermore, there are 9 diagnostic ions in the MS/MS spectrum with 2 of these being backbone fragments, thus also meets the identification criteria.

31. RvD5n-3DPA. Very poor quality peak, that would not pass criteria. The spectrum is not good enough to confirm the lipid, with baseline ions labeled as diagnostic ions and no ion at 199 or 201, despite labels being included. 143 is also missing. The lipid is judged to be absent from the sample.

It is unclear to us how the reviewer determined that this peak would not pass criteria given that as can be observed in Supplemental Figure 2G and Supplemental Table 2 the peak presented meets the criteria outlined by the reviewer in their comment above in that the retention time is a match with that of the standard, the peak is composed of 6 data points and the S/N ratio is of 12. Furthermore, the ions that the reviewer states are missing are in the MS/MS spectrum are clearly labeled. In regards to the intensity of the 143 fragment the reviewer is referred to the reference spectrum provided.

32. PD1n-3DPA. Baseline noise in a chromatogram being labeled as a peak. Spectrum does not match the spectrum in the handbook, missing significant ions at 155, 183, 263. The lipid is judged to be absent from the sample.

We are in disagreement given that as can be observed in Supplemental Figure 2H and

Supplemental Table 2 the peak presented meets the criteria outlined by the reviewer in their comment above in that the retention time is a match with that of the standard, the peak is composed of 5 data points and the S/N ratio is of 12. Furthermore, the ions that the reviewer states are missing in the MS/MS spectrum are clearly labelled in the reference spectrum provided.

33. MaR1n-3DPA. Baseline noise in a chromatogram being labeled as a peak. Spectrum does not match the spectrum in the handbook, missing diagnostic ions (they look like background noise at 143, 223, 233). The lipid is judged to be absent from the sample.

We respectfully disagree, given that as can be observed in Supplemental Figure 2I and Supplemental Table 2 the peak presented meets the criteria outlined by the reviewer in their comment above in that the retention time is a match with that of the standard, the peak is composed of 11 data points and the S/N ratio is of 13. Furthermore, as discussed above, due to matrix contribution ion fragmentation profiles change. Nonetheless, the MS/MS spectrum contains 9 diagnostic fragments of which 2 are direct backbone fragments and therefore fulfills the identification criteria.

34. RvE1. A weak present is present, around same RT as in standard handbook, however, spectrum is very different to handbook. To be sure this lipid was the correct one, the standard should be run under the same conditions on the same column (this information is not provided). Given how small the peak is it is, it is hard to be convinced this is really properly identified. Borderline but possibly present.

We have provided the reference spectrum obtained under identical conditions and in the presence of matrix. As can be appreciated from results presented in Supplemental Figure 3A and Supplemental Table 2 both the peak (matching retention time, the peak is composed of 7 data points and the S/N ratio is of 8) and MS/MS fragmentation spectrum (9 diagnostic ions, 1 of which are direct backbone fragments) fulfill identification criteria.

35. RvE2. A weak present is present, around same RT as in standard handbook, however, spectrum is very different to handbook, missing ions at 115, 213. To be sure this lipid was the correct one, the standard should be run under the same conditions on the same column (this information is not provided). Given how small the peak is it is, it is hard to be convinced this is really properly identified. Borderline but possibly present.

We have provided the reference spectrum obtained under identical conditions and in the presence of matrix. As can be appreciated from results presented in Supplemental Figure 3B and Supplemental Table 2 both the peak (matching retention time, the peak is composed of 9 data points and the S/N ratio is of 24) and MS/MS fragmentation spectrum (8 diagnostic ions, 1 of which are backbone fragments) fulfill identification criteria.

36. RvE3. Baseline noise on the side of a noisy peak, which may not be a peak itself is being

integrated. Spectrum is not convincingly like the standard. The lipid is judged to be absent from the sample.

We are uncertain what noise is being referred to in this comment. As previously discussed many of the mediators have natural isomers that elute close to the bioactive mediators, however these should not be regarded as noise. In addition, as can be observed in Supplemental Figure 3C and Supplemental Table 2, the chromatogram fulfills the identification criteria in that the retention time is a match with that of the standard, the peak is composed of 5 data points and the S/N ratio is of 5. Furthermore, there are 7 ions in the MS/MS spectrum, one of which is a direct backbone fragment. Therefore, it is unclear why the reviewer states that this is not convincing.

37. LXA4. No peak in this chromatogram, just baseline noise. A weak spectrum that is not helping with identification. The lipid is judged to be absent from the sample.

We agree with the reviewer unfortunately we included the wrong chromatogram in the figure. This mistake has been rectified as can be observed in Supplemental Figure 4A and Supplemental Table 2. As can be observed from the chromatogram the peak fulfills the identification criteria in that the retention time is a match with that of the standard, the peak is composed of 5 data points and the S/N ratio is of 7. The MS/MS spectrum is composed of 8 diagnostic ions 2 of which are direct backbone fragments, this confirming the presence of the mediator.

38. LXB4. A peak that looks ok, but is not particularly strong. Spectrum has some diagnostic ions. Presuming that this lipid co-elutes with the standard, then it should be the correct one.

As can be observed from Supplemental Figure 4B and Supplemental Table 1 the mediator does indeed co-elute with the standard.

39. 5S,15S-diHETE. A weak shoulder alongside a bigger peak that doesn't pass quality, and is surrounded by several other similar weak peaks. The spectrum has blown up a lot of noise and so deciding that these are diagnostic signals is impossible. Some larger ions that would not be expected (195, 165) are ignored for example. The lipid judged absent.

As previously discussed, the presence of other peaks is typical for lipid mediators given that there are many natural isomers to these molecules and the MRM transitions are not unique to a single molecule as is common for drug metabolites for example. As can be appreciated in Supplemental Figure 4C and Supplemental Table 2 the identified peak does meet the identification criteria in that the retention time is a match with that of the standard, the peak is composed of 4 data points and the S/N ratio is of 7. The MS/MS spectrum presents with 10 diagnostic ions including 2 backbone fragments we deem this a positive match. The unlabeled ions referred to by the reviewer may result from interactions of the molecule with the matrix, however since these we were unable to confidently assign them to a specific portion of the molecule therefore we chose to exclude them from the list of ions used for identification.

40. 15-epi-LXA4. Peak is background noise and does not show the lipid to be present. A weak and unconvincing spectrum. The lipid is judged to be absent from the sample.

We are in disagreement given that as can be observed in Supplemental Figure 4D and Supplemental Table 2 the peak presented meets the criteria outlined by the reviewer in their comment above in that the retention time is a match with that of the standard, the peak is composed of 8 data points and the S/N ratio is of 8. Furthermore, there are 9 diagnostic ions including 4 backbone fragments in the MS/MS spectrum, thus confirming the presence of this mediator in the sample

41. 15-epi-LXB4. Peak is background noise and does not show the lipid to be present. A weak and unconvincing spectrum. The lipid is judged to be absent from the sample.

We are in disagreement given that as can be observed in Supplemental Figure 4E and Supplemental Table 2 the peak presented meets the criteria outlined by the reviewer in their comment above in that the retention time is a match with that of the standard, the peak is composed of 8 data points and the S/N ratio is of 5. Furthermore, there are 9 diagnostic ions including 3 backbone fragments in the MS/MS spectrum, thus confirming the presence of this mediator in the sample

42. LTB4: The chromatogram and spectrum are convincingly showing the lipid
OK

43. 5S,12S-diHETE: this is noise in the baseline, and is shown as a tiny shoulder attached to a bigger peak. It would be impossible to get a spectrum of this shoulder without the larger adjacent peak being also fragmented. The lipid is judged to be absent from the sample

We are in disagreement, the larger peak referred to by the reviewer is that of LTB4, its natural isomer that elutes very closely. Furthermore, given as can be observed in Supplemental Figure 4G and Supplemental Table 2 the peak presented meets the criteria outlined by the reviewer in their comment above in that the retention time is a match with that of the standard, the peak is composed of 4 data points and the S/N ratio is of 7.

44. 6-trans-12-epi-LTB4. This is a peak presumably measured in the same channel as LTB4 (unclear), and it would be essential to see a standard run alongside so that RT can be confirmed (this was not shown). No spectrum is available, but comparing with LTB4 there are some similarities. Insufficient information provided but convincing peak shown.

We provided a reference chromatogram and MS/MS spectrum, and as can be appreciated the retention time and fragment ions in the MS/MS spectrum depicted for this molecule in the sample match those for the reference standard as can be observed in Supplemental Figure 4I and Supplemental Table 2.

45. 6-trans-LTB4. Potentially there, although needs to be shown alongside standard on chromatogram for retention time. Spectrum is not showing the lipid convincingly. Insufficient information provided but possibly present as peak is convincing.

We provided a reference chromatogram and MS/MS spectrum, and as can be appreciated in Supplemental Figure 4H and Supplemental Table 2 the retention time and fragment ions in the MS/MS spectrum depicted for this molecule in the sample match those for the reference standard.

46. 20-COOH-LTB4. A decent peak, and spectrum looks similar, so possible match although the retention time is very different from the handbook at 0.6min later, so the RT of the standard needs to be provided. Insufficient information provided but potential match as strong peak shown.

We provided a reference chromatogram and MS/MS spectrum (Supplemental Figure 4K and Supplemental Table 2), and as can be appreciated the retention time and fragment ions in the MS/MS spectrum depicted for this molecule in the sample match those for the reference standard.

47. LTC4. Retention time is a bit early so wonder if the right peak has been integrated. Should be around 10.3, and there is another peak there. But the peak is ok as shown. Spectrum appears reasonable, so probably ok.

We provided a reference chromatogram and MS/MS spectrum (Supplemental Figure 4J and Supplemental Table 2), and as can be appreciated the retention time and fragment ions in the MS/MS spectrum depicted for this molecule in the sample match those for the reference standard.

48. LTD4. Chromatogram and spectrum are fine

OK

49. LTE4. Chromatogram and spectrum are fine

OK

50. PGD2. Chromatogram and spectrum are fine

OK

51. PGE2. Chromatogram and spectrum are fine

OK

52. PGF2a. Peak is low and borderline, but I suspect this one is ok

OK

53. TXB₂. This normally elutes as a sort of blob or double peak, and I suspect the whole area should have been integrated. Instead they have taken a small shoulder at the start and ignored the rest. The lipid is likely to be there, but the quantitation is poor and missing most of the lipid

The reviewer is correct, the chromatography of this molecule is widely appreciated to be difficult. Given that the entire peak for TXB₂ is not captured in our MRM transition, in order to ensure that the quantitation is robust and reproducible we have opted to integrate only the initial portion of the peak based on peak width which we find gives us reproducible results for quantitation purposes. This portion of the peak is integrated in both the standard and samples therefore making the quantitation correct, as shown Supplemental Figure 4R and Supplemental Table 2.

Reviewer #5 (Remarks to the Author):

Comments on the lipid analyses by LC-MS

Criteria for the positive identification and quantitation of analytes by LC-MS include among other parameters elution at the correct retention time, a “clean-looking” LC peak not co-eluting with nearby compounds, and an acceptable level of signal-to-noise.

The chromatograms illustrated in Supplement Figure A - X and A - I include many analyses that do not achieve the required uncontaminated appearance of the chromatographic peak and many cases in which the peaks exhibit a very low and unacceptable signal-to-noise ratio.

What is an acceptable signal-to-noise ratio? This number should be defined in the methods such that a signal-to-noise below the cut-off be rejected and the analysis deemed invalid. One might suggest that a low (non-stringent) cut-off be, for example, 5:1 signal:noise. Many of the representative chromatograms in the Supplement do not achieve even such a non-stringent criterion and very clearly should be rejected in any generally acceptable and validated LC-MS methodology.

We have now included the relevant signal to noise information together with the number of data points for each of the chromatograms used. This information can be found in Supplemental Figures 1-4 and summarized in supplemental Table 2. As can be observed from the results presented the signal to noise ratio for all of the mediators identified is of 4:1 or greater. We also would like to note that 2 of the representative chromatograms that were included in the previous data set were included erroneously and we have endeavored to rectify this mistake.

We however disagree with the idea that the peak has to be completely distinct, a comment that we interpret as having to have baseline separation between the peaks. The reason for our disagreement is because many of the lipoxygenase-derived mediators have natural isomers which elute very close to each other and therefore for such molecules there are always going to be biologically meaningful peaks that elute before and/or after the one of interest. Of note, this is no different to many of the classic eicosanoids where for example PGD₂ and PGE₂ elute very closely as do LTB₄, 5S, 12S-diHETE, and Δ6trans, 12-epi-LTB₄, which as noted by Reviewer 4 are quantified by many groups around the world.

In each case in the relevant Supplement chromatograms the investigators include a mass spectrum (or product ion spectrum) that gives some support to the designated peak identity.

However, this cannot overrule a contaminated chromatographic peak and/or an unacceptably low signal-to-noise ratio. When the signal-to-noise ratio is 1:1 or near that, one would question what is the appearance of the mass spectra in nearby areas of the chromatogram, (the illustrated chromatograms are already selected for the designated molecular ion)? Also, to what extent do the mass spectra fragmentations exactly match the standard – what leeway is taken in this parameter?

The mass spectra are obtained from the same area where the molecule of interest is observed to elute to minimize the likelihood of false positive identification.

[Minor, typo: Main manuscript page 17, 4th line from bottom: It appears unlikely the retention time match could be restricted to 0.05 sec as stated].

This is not a typo and indeed the retention time drift from standard is restricted to 0.05 seconds.

Reviewers' comments:

Reviewer #6 (Remarks to the Author):

This mediation reviewer is of the opinion that some of the criticisms raised by previous reviewers are and remain valid. Without going into specific details of each analyte results presented, the following overarching considerations and questions remain unaddressed and unanswered.

What precisely is shown in Supplementary Figure 1 (left panels)? These are chromatograms but what is the underlying mass spectrometric measurement? Or in other words, what are the MRM transitions used for quantification of lipids? Reference is made to publication 35 which leads to reference 22 followed by reference 28. However, none of these previous papers provide much information on instrument parameters used for quantification by mass spectrometry, in particular MRM transitions.

The data presented in Supplemental Figure 1 changed considerably during the most recent revision (eg. RvD2 or RvD4 where alternative chromatograms are shown now). This would indicate that chromatograms vary extensively from run to run. It would therefore be critical to include data on experimental, ie. technical variability (such as coefficients of variations, CoV). However, the technical variability of the measurements performed here is not mentioned in the manuscript. Determination and tracking of the technical variability is a central element of quality assurance and control in analytical chemistry. Cut-offs values in the range of 5-25% (depending on the application and stringency of a particular test) are used, above which results from an experiment or a set of runs would be excluded. In mass spectrometry-based lipidomics an often-accepted limit is <25%. Technical CoV has an inherent, non-linear, dependency on signal intensity. It therefore needs to be determined for each analyte and condition separately. The lower the signal, the higher the variability. Therefore, this becomes particularly relevant for low abundant metabolites such as the ones of interest here. There are various ways to estimate technical variabilities. The authors should be able to do so one way or another and provide this information. Knowledge of technical variability is fundamental for any discovery and measurement of biological variability. No credible claims can be made if the former is equal or larger than the latter.

How is 'noise' defined and determined in these experiments? This could be a constructive suggestion for potential mediation of arguments between the authors and reviewer 4 in particular. Several approaches are commonly used to determine noise levels in a mass spectrometric measurement. First, it can be estimated from the baseline values before and after a peak/signal in a chromatogram. Instrument-specific software provided by the equipment manufacturer most often provide such information, although the precise algorithms for noise determination might differ from vendor-to-vendor and between models. Alternatively, and also commonly utilized, are blank or solvent extracts. These are convenient ways to determine the baseline noise level using otherwise identical analytical conditions. Finally, blank, stripped matrix or synthetic matrix devoid of analyte is an ideal control measure which however is difficult to obtain and therefore not commonly used in lipidomics. Defining noise (the denominator in the S/N equation) seems like a sensible way to break the apparent circular arguments between the authors and reviewer 4.

Reviewer #6 (Remarks to the Author):

This mediation reviewer is of the opinion that some of the criticisms raised by previous reviewers are and remain valid. Without going into specific details of each analyte results presented, the following overarching considerations and questions remain unaddressed and unanswered.

We thank the reviewer for their input on our manuscript and for their efforts in mediating our response.

What precisely is shown in Supplementary Figure 1 (left panels)? These are chromatograms but what is the underlying mass spectrometric measurement? Or in other words, what are the MRM transitions used for quantification of lipids? Reference is made to publication 35 which leads to reference 22 followed by reference 28. However, none of these previous papers provide much information on instrument parameters used for quantification by mass spectrometry, in particular MRM transitions.

We have added the relevant information of MRM transitions used for each of the chromatograms presented in supplemental figures 1-4. We also added this information to Supplementary Tables 3, 5, 7 and have revised the reference provided in the methods (see references 20 and 34).

The data presented in Supplemental Figure 1 changed considerably during the most recent revision (eg. RvD2 or RvD4 where alternative chromatograms are shown now). This would indicate that chromatograms vary extensively from run to run. It would therefore be critical to include data on experimental, ie. technical variability (such as coefficients of variations, CoV). However, the technical variability of the measurements performed here is not mentioned in the manuscript. Determination and tracking of the technical variability is a central element of quality assurance and control in analytical chemistry. Cut-offs values in the range of 5-25% (depending on the application and stringency of a particular test) are used, above which results from an experiment or a set of runs would be excluded. In mass spectrometry-based lipidomics an often-accepted limit is <25%. Technical CoV has an inherent, non-linear, dependency on signal intensity. It therefore needs to be determined for each analyte and condition separately. The lower the signal, the higher the variability. Therefore, this becomes particularly relevant for low abundant metabolites such as the ones of interest here. There are various ways to estimate technical variabilities. The authors should be able to do so one way or another and provide this information. Knowledge of technical variability is fundamental for any discovery and measurement of biological variability. No credible claims can be made if the former is equal or larger than the latter.

We respectfully disagree with the comment that the data changed significantly. Where aspects of the data were substituted or modified for clarity purposes we highlighted this in our prior response. In the examples mentioned we explained that the chromatogram for RvD2 was changed simply for clarity reasons and that the peak in the chromatogram used before also fulfilled the criteria. As for RvD4 we acknowledge that the chromatogram initially provided was not the correct one and we therefore corrected this mistake by providing the right chromatogram. It should also be noted that variability in chromatographic appearance of specific transitions is an inherent phenomenon in the measurement of bioactive mediators. This is because the concentrations of natural isomers of these molecules are known to change from sample to sample and individual to individual therefore changing the overall appearance of the chromatogram.

The reviewer is correct that determining the technical variability of the method is essential and we have conducted and published such experiments in the past, we have included a

reference to this in our manuscript (reference 34). Nonetheless, we have included the values for assay variability on the instrumentation that was used to acquire the data for the present manuscript. This information is included in Supplemental Tables 11 and 12 of the revised manuscript. As can be observed from these results the variability for each of the transitions employed to quantify the mediators at concentration ranges within those reported in the present manuscript, i.e. down to 0.5pg, were less than the 25% cut-off indicated by the reviewer.

How is 'noise' defined and determined in these experiments? This could be a constructive suggestion for potential mediation of arguments between the authors and reviewer 4 in particular. Several approaches are commonly used to determine noise levels in a mass spectrometric measurement. First, it can be estimated from the baseline values before and after a peak/signal in a chromatogram. Instrument-specific software provided by the equipment manufacturer most often provide such information, although the precise algorithms for noise determination might differ from vendor-to-vendor and between models. Alternatively, and also commonly utilized, are blank or solvent extracts. These are convenient ways to determine the baseline noise level using otherwise identical analytical conditions. Finally, blank, stripped matrix or synthetic matrix devoid of analyte is an ideal control measure which however is difficult to obtain and therefore not commonly used in lipidomics. Defining noise (the denominator in the S/N equation) seems like a sensible way to break the apparent circular arguments between the authors and reviewer 4.

Thank you for the suggestions. Noise in the present study was determined on the region before or after the peak of interest in the chromatogram. Here we selected the closest possible region to the peak of interest considering any biological isomers in this region. The region designated as the noise in our S/N calculation has been highlighted for each of the chromatograms displayed in Supplemental Figures 1-4. After also highlighting the peak of interest, we used the S/N tool available in Analyst 1.6.3 to determine the signal to noise value. Here the application tool divides the intensity value for the region denoted as the signal/peak of interest by the intensity value for the highest peak in the region denoted as noise.

REVIEWER COMMENTS

Reviewer #6 (Remarks to the Author):

Original Comment:

This mediation reviewer is of the opinion that some of the criticisms raised by previous reviewers are and remain valid. Without going into specific details of each analyte results presented, the following overarching considerations and questions remain unaddressed and unanswered.

Author Response:

We thank the reviewer for their input on our manuscript and for their efforts in mediating our response.

Original Comment:

What precisely is shown in Supplementary Figure 1 (left panels)? These are chromatograms but what is the underlying mass spectrometric measurement? Or in other words, what are the MRM transitions used for quantification of lipids? Reference is made to publication 35 which leads to reference 22 followed by reference 28. However, none of these previous papers provide much information on instrument parameters used for quantification by mass spectrometry, in particular MRM transitions.

Author Response:

We have added the relevant information of MRM transitions used for each of the chromatograms presented in supplemental figures 1-4. We also added this information to Supplementary Tables 3, 5, 7 and have revised the reference provided in the methods (see references 20 and 34).

New Reviewer Comment:

The authors have made progressive advancements in the documentation of the analytical results during the successive stages of revisions. These included addition of chromatograms, clarification that these are indeed MRM chromatograms, and the transitions used to monitor each analyte. Data for both standards in matrix as well as samples are now shown for a large number of analytes including analytical criteria used for identification and quantification (e.g. retention time, S/N ratio, number of data points per peak). Integration areas as well as regions utilized for determination of the noise levels are now also clearly indicated for each analyte. These revisions have improved the manuscript considerably on accounts of clarity and reporting standards of raw data.

Original Comment:

The data presented in Supplemental Figure 1 changed considerably during the most recent revision (eg. RvD2 or RvD4 where alternative chromatograms are shown now). This would indicate that chromatograms vary extensively from run to run. It would therefore be critical to include data on experimental, ie. technical variability (such as coefficients of variations, CoV). However, the technical variability of the measurements performed here is not mentioned in the manuscript. Determination and tracking of the technical variability is a central element of quality assurance and control in analytical chemistry. Cut-offs values in the range of 5-25% (depending on the application and stringency of a particular test) are used, above which results from an experiment or a set of runs would be excluded. In mass spectrometry-based lipidomics an often-accepted limit is <25%. Technical CoV has an inherent, non-linear, dependency on signal intensity. It therefore needs to be determined for each analyte and condition separately. The lower the signal, the higher the variability. Therefore, this becomes particularly relevant for low abundant metabolites such as the ones of interest here. There are various ways to estimate technical variabilities. The authors should be able to do so one way or another and provide this information. Knowledge of technical variability is fundamental for any discovery and measurement of biological variability. No credible claims can be made if the former is equal or larger than the latter.

Author Response:

We respectfully disagree with the comment that the data changed significantly. Where aspects of the data were substituted or modified for clarity purposes we highlighted this in our prior response. In the examples mentioned we explained that the chromatogram for RvD2 was changed simply for clarity reasons and that the peak in the chromatogram used before also fulfilled the criteria. As for RvD4 we acknowledge that the chromatogram initially provided was not the correct one and we therefore corrected this mistake by providing the right chromatogram. It should also be noted that variability in chromatographic appearance of specific transitions is an inherent phenomenon in the measurement of bioactive mediators. This is because the concentrations of natural isomers of these molecules are known to change from sample to sample and individual to individual therefore changing the overall appearance of the chromatogram.

New Reviewer Comment:

This was my point. The change in overall appearance of the chromatogram from sample to sample and run to run and how this relates to quantification of the biological effects.

Author Response:

The reviewer is correct that determining the technical variability of the method is essential and we have conducted and published such experiments in the past, we have included a reference to this in our manuscript (reference 34). Nonetheless, we have included the values for assay variability on the instrumentation that was used to acquire the data for the present manuscript. This information is included in Supplemental Tables 11 and 12 of the revised manuscript. As can be observed from these results the variability for each of the transitions employed to quantify the mediators at concentration ranges within those reported in the present manuscript, i.e. down to 0.5pg, were less than the 25% cut-off indicated by the reviewer.

New Reviewer Comment:

The variability values in Supplemental Tables 11 and 12 are indeed remarkably low given the visible levels of noise in many of the chromatograms shown in Supplemental Figure 1-4.

Original Comment:

How is 'noise' defined and determined in these experiments? This could be a constructive suggestion for potential mediation of arguments between the authors and reviewer 4 in particular. Several approaches are commonly used to determine noise levels in a mass spectrometric measurement. First, it can be estimated from the baseline values before and after a peak/signal in a chromatogram. Instrument-specific software provided by the equipment manufacturer most often provide such information, although the precise algorithms for noise determination might differ from vendor-to-vendor and between models. Alternatively, and also commonly utilized, are blank or solvent extracts. These are convenient ways to determine the baseline noise level using otherwise identical analytical conditions. Finally, blank, stripped matrix or synthetic matrix devoid of analyte is an ideal control measure which however is difficult to obtain and therefore not commonly used in lipidomics. Defining noise (the denominator in the S/N equation) seems like a sensible way to break the apparent circular arguments between the authors and reviewer 4.

Author Response:

Thank you for the suggestions. Noise in the present study was determined on the region before or after the peak of interest in the chromatogram. Here we selected the closest possible region to the peak of interest considering any biological isomers in this region. The region designated as the noise in our S/N calculation has been highlighted for each of the chromatograms displayed in Supplemental Figures 1-4. After also highlighting the peak of interest, we used the S/N tool available in Analyst 1.6.3 to determine the signal to noise value. Here the application tool divides the intensity value for the region denoted as the signal/peak of interest by the intensity value for the highest peak in the region denoted as noise.

New Reviewer Comment:

Brackets used for estimation of noise are now shown for all analytes (green bands in supplemental

figures 1-4). Inclusion of this information is helpful for interpretation of the raw data with respect to noise vs signal. This reviewer has identified the noise aspect as central for mediation between the authors and previous reviewers, in particular reviewer 4.

Two additional points should be further clarified in that respect:

1. Widths of the noise brackets

All areas are narrower than those used for integration of the analyte. Similar chromatographic widths should be used instead.

2. Positions of noise windows

What determines the precise placement of the noise level brackets? In some cases, estimates were taken immediately before or after the elution time of an analyte of interest. It is stated above that potential presence of biological isomers somehow guided this choice.

However, and importantly, how does the overall quantification outcome and conclusion of this study depend on when precisely noise levels are determined during the chromatographic elution? Increased values for the denominator N will lower the S/N ratios for a selected signal S. Based on visual inspection of the chromatograms it would not be surprising if the noise level estimates are sensitive to slight re-positioning of the green window.

It is therefore quite possible that these two factors, i.e. widths and positions of noise windows, could affect the outcome for a number of mediators, in particular if $S/N < 4$ is used as an exclusion criterion as mentioned in rebuttal to comments by reviewer 4. Many analytes have $S/N < 10$, including all of the four mentioned in the abstract.

NCOMMS-19-17846C **New Reviewer Comment:**

Brackets used for estimation of noise are now shown for all analytes (green bands in supplemental figures 1-4). Inclusion of this information is helpful for interpretation of the raw data with respect to noise vs signal. This reviewer has identified the noise aspect as central for mediation between the authors and previous reviewers, in particular reviewer 4.

Two additional points should be further clarified in that respect:

1. Widths of the noise brackets

All areas are narrower than those used for integration of the analyte. Similar chromatographic widths should be used instead.

We thank the reviewer for their response. The reviewer is correct that the area is narrower than that of the peak of interest. We would like to point out that, to best of our knowledge, there is no consensus on the width and location of the region to be identified as noise in the literature, a topic that is also discussed in the literature referenced here where several approaches are proposed¹⁻⁴. In our analysis we defined the noise region that was commensurate with our chromatographic method and the ability to separate biological isomers. Nonetheless, we repeated the analysis using area equal to that of the peak of interest as suggested by the reviewer. The revised figures are now included in supplemental figure 1-4 of the manuscript. As can be observed from the results obtained in the table appended below, overall the S/N values are not markedly changed when using a different noise area.

2. Positions of noise windows

What determines the precise placement of the noise level brackets? In some cases, estimates were taken immediately before or after the elution time of an analyte of interest. It is stated above that potential presence of biological isomers somehow guided this choice.

However, and importantly, how does the overall quantification outcome and conclusion of this study depend on when precisely noise levels are determined during the chromatographic elution? Increased values for the denominator N will lower the S/N ratios for a selected signal S. Based on visual inspection of the chromatograms it would not be surprising if the noise level estimates are sensitive to slight re-positioning of the green window.

It is therefore quite possible that these two factors, i.e. widths and positions of noise windows, could affect the outcome for a number of mediators, in particular if $S/N < 4$ is used as an exclusion criterion as mentioned in rebuttal to comments by reviewer 4. Many analytes have $S/N < 10$, including all of the four mentioned in the abstract.

The reviewer is correct in their observation that the location of the area designated as noise in the S/N calculation is influenced by the number of biological isomers and the proximity that these isomers elute. This unavoidable given that for many of the molecules of interest there are a number of naturally occurring biological isomers that elute close to the peak of interest.

The reviewer is also correct that the S/N measure is sensitive to the region that is used with the chromatogram, this is an aspect that is also well appreciated in the field. To the best of our knowledge, there is no consensus in the literature establishing what the proximity of the region to be used as noise should be in relation to the peak of interest¹⁻⁴.

Similarly, there is also no consensus in the literature on what the minimum acceptable S/N values should be, with values as low as S/N of 2-3 being indicated^{1,2}, all the way up to an S/N of 100. Furthermore, and as detailed by the reviewer, there are multiple ways that background noise can be determined, and each of these approaches will give somewhat different results. Given all of these aspects, as detailed in the manuscript, our routine identification criteria, which are extensively published⁵⁻¹³, rely on the number of data points that contribute to the peak and a minimum peak area. These chromatographic parameters are further supported by MS/MS spectral data. This aspect was also discussed in response to reviewer 4 and the inclusion of the S/N parameter was to provide a comparison of our criteria with those detailed by the reviewer. This comparison demonstrated that there is significant overlap between the criteria we employed and those indicated by reviewer 4 for chromatographic identification of a molecule. Therefore, while consideration is given to the S/N value for a given molecule, it does not form part of our core identification criteria. Thus, this parameter has does not impact on the overall quantification outcome and conclusions of this study.

References:

1. Zhang J, Gonzalez E, Hestilow T, Haskins W, Huang Y. Review of peak detection algorithms in liquid-chromatography-mass spectrometry. *Curr Genomics* 2009; **10**(6): 388-401.
2. TL S, RA Y. What's the Most Meaningful Standard for Mass Spectrometry : Instrument Detection Limit or Signal-to-Noise Ratio ? *Spectroscopy* 2017.
3. Evard H, Krueve A, Leito I. Tutorial on estimating the limit of detection using LC-MS analysis, part I: Theoretical review. *Anal Chim Acta* 2016; **942**: 23-39.
4. https://ec.europa.eu/food/sites/food/files/safety/docs/animal-feed-guidance_document_lod_en.pdf.
5. Arnardottir HH, Dalli J, Norling LV, Colas RA, Perretti M, Serhan CN. Resolvin D3 Is Dysregulated in Arthritis and Reduces Arthritic Inflammation. *J Immunol* 2016; **197**(6): 2362-8.
6. Chiang N, Dalli J, Colas RA, Serhan CN. Identification of resolvin D2 receptor mediating resolution of infections and organ protection. *J Exp Med* 2015; **212**(8): 1203-17.
7. Colas RA, Shinohara M, Dalli J, Chiang N, Serhan CN. Identification and signature profiles for pro-resolving and inflammatory lipid mediators in human tissue. *Am J Physiol Cell Physiol* 2014; **307**(1): C39-54.
8. Dalli J, Chiang N, Serhan CN. Elucidation of novel 13-series resolvins that increase with atorvastatin and clear infections. *Nat Med* 2015; **21**(9): 1071-5.
9. Dalli J, Colas RA, Walker ME, Serhan CN. Lipid Mediator Metabolomics Via LC-MS/MS Profiling and Analysis. *Methods Mol Biol* 2018; **1730**: 59-72.
10. Dalli J, Ramon S, Norris PC, Colas RA, Serhan CN. Novel proresolving and tissue-regenerative resolvin and protectin sulfido-conjugated pathways. *FASEB J* 2015; **29**(5): 2120-36.
11. Fredman G, Ozcan L, Spolitu S, et al. Resolvin D1 limits 5-lipoxygenase nuclear localization and leukotriene B4 synthesis by inhibiting a calcium-activated kinase pathway. *Proc Natl Acad Sci U S A* 2014; **111**(40): 14530-5.
12. Freire MO, Dalli J, Serhan CN, Van Dyke TE. Neutrophil Resolvin E1 Receptor Expression and Function in Type 2 Diabetes. *J Immunol* 2017; **198**(2): 718-28.

13. Norling LV, Headland SE, Dalli J, et al. Proresolving and cartilage-protective actions of resolvin D1 in inflammatory arthritis. *JCI Insight* 2016; **1**(5): e85922.

Table 1: Comparison of S/N values

	S/N ratio (Original calculated)	S/N ratio (Noise region same width of analyte peak)
DHA Bioactive Metabolome		
RvD1	12	12
RvD2	8	8
RvD3	6	5
RvD4	9	7
RvD5	6	5
RvD6	11	5
17R-RvD1	6	4
17R-RvD3	5	3
PD1	5	5
17R-PD1	5	6
PDx	7	4
22-OH-PD1	31	31
PCTR1	7	4
PCTR2	6	6
PCTR3	12	7
MaR1	5	5
MaR2	30	11
22-OH-MaR1	6	4
14-oxo-MaR1	23	24
7S,14S-diHDHA	23	5
4,14-diHDHA	11	9
MCTR1	17	14
MCTR2	5	5
MCTR3	13	13
n-3 DPA bioactive Metabolole		
RvT1	13	10
RvT2	6	6
RvT3	8	8
RvT4	7	7
RvD1 _{n-3 DPA}	11	13
RvD2 _{n-3DPA}	10	4
RvD5 _{n-3DPA}	12	6
PD1 _{n-3 DPA}	11	6
10S,17S-diHPDA	5	4
MaR1 _{n-3 DPA}	5	5
EPA bioactive Metabolome		
RvE1	16	6
RvE2	24	8
RvE3	5	8
AA bioactive Metabolome		
LXA ₄	5	4
LXB ₄	7	11
5S,15S-diHETE	7	7
15R-LXA ₄	8	8
15R-LXB ₄	5	4
LTB ₄	69	37
5S,12S-diHETE	7	7
6-trans-LTB ₄	19	34
12-epi-6-trans-LTB ₄	28	48
20-OH-LTB ₄	88	38
20-COOH-LTB ₄	60	24
LTC ₄	17	17
LTD ₄	30	30
LTE ₄	15	33
PGD ₂	55	41

PGE₂
PGF_{2a}
TxB₂

39	29
8	8
8	8

REVIEWERS' COMMENTS

Reviewer #6 (Remarks to the Author):

The signal to noise ratios were reduced for approximately 50% of the analytes by widening of the noise window. For close to 20% of analytes it dropped to half of their previous values. Thus, it is not quite correct to state that the signal to noise ratios are not markedly changed. Two out of the four molecules mentioned in the abstract are now at or slightly above the signal to noise ratio of 4 used as an inclusion criterion. But with this revision the authors have further added transparency to their method which is valuable to the reader and now quite well documented here.

Point by Point Response to Reviewer Comments

Reviewer #6 (Remarks to the Author):

The signal to noise ratios were reduced for approximately 50% of the analytes by widening of the noise window. For close to 20% of analytes it dropped to half of their previous values. Thus, it is not quite correct to state that the signal to noise ratios are not markedly changed. Two out of the four molecules mentioned in the abstract are now at or slightly above the signal to noise ratio of 4 used as an inclusion criterion. But with this revision the authors have further added transparency to their method which is valuable to the reader and now quite well documented here.

Thank you for the time taken to mediate our manuscript and the insightful comments provide. We are glad to learn that the reviewer finds the revisions made suitable to address the comments raised.